# Unified Generative and Discriminative Training for Multi-modal Large Language Models

**Wei Chow**[1]    **Juncheng Li**[1,†]    **Qifan Yu**[1]    **Kaihang Pan**[1]    **Hao Fei**[2]
**Zhiqi Ge**[1]    **Shuai Yang**[1]    **Siliang Tang**[1,†]    **Hanwang Zhang**[3]    **Qianru Sun**[4]
[1]Zhejiang University    [2]National University of Singapore
[3]Nanyang Technological University    [4]Singapore Management University
{xieqiao, junchengli, yuqifan, kaihangpan}@zju.edu.cn
{zhiqige, syang, siliang}@zju.edu.cn
haofei37@nus.edu.sg, hanwangzhang@ntu.edu.sg, qianrusun@smu.edu.sg

## Abstract

In recent times, Vision-Language Models (VLMs) have been trained under two predominant paradigms. Generative training has enabled Multimodal Large Language Models (MLLMs) to tackle various complex tasks, yet issues such as hallucinations and weak object discrimination persist. Discriminative training, exemplified by models like CLIP, excels in zero-shot image-text classification and retrieval, yet struggles with complex scenarios requiring fine-grained semantic differentiation. This paper addresses these challenges by proposing a unified approach that integrates the strengths of both paradigms. Considering interleaved image-text sequences as the general format of input samples, we introduce a structure-induced training strategy that imposes semantic relationships between input samples and the MLLM's hidden state. This approach enhances the MLLM's ability to capture global semantics and distinguish fine-grained semantics. By leveraging dynamic sequence alignment within the Dynamic Time Warping framework and integrating a novel kernel for fine-grained semantic differentiation, our method effectively balances generative and discriminative tasks. Extensive experiments demonstrate the effectiveness of our approach, achieving state-of-the-art results in multiple generative tasks, especially those requiring cognitive and discrimination abilities. Additionally, our method surpasses discriminative benchmarks in interleaved and fine-grained retrieval tasks. By employing a retrieval-augmented generation strategy, our approach further enhances performance in some generative tasks within one model, offering a promising direction for future research in vision-language modeling. The project repository is here.

## 1 Introduction

In recent times, Vision-Language Models (VLMs) have been trained under two predominant paradigms: generative training and discriminative training. **Generative Training** has achieved remarkable success in enabling Multimodal Large Language Models (MLLMs) [1, 55, 86] to develop a wide range of powerful capabilities that can handle various complex tasks (*e.g.,* open-world visual question-answering, image caption generation, etc.) within a single model. However, challenges such as hallucinations and weak image object discrimination abilities [7, 89] persist. **Discriminative Training**, exemplified by CLIP [73], exhibits remarkable representation capabilities for zero-shot image-text classification and retrieval. Nonetheless, it encounters difficulties in processing complex scenarios (*i.e.,* , retrieving multi-modal documents with interleaved images and texts) [53, 54] and exhibits a limited ability to discern detailed semantic differences [79, 85].

---

† Corresponding Author.

38th Conference on Neural Information Processing Systems (NeurIPS 2024).

The disparity between these two paradigms has sparked recent studies aimed at imparting discriminative ability to generative pre-trained MLLMs. However, certain aspects of performance still pose limitations (*e.g.,* singular discriminative tasks [89], weak discriminative task performance [40], weak generalization [59], etc.), while others entail compromising the model's original generative capabilities [8].

Overall, the reason generative paradigms struggle with performing discriminative tasks like retrieval is due to overlooking two crucial abilities:

*(i)* **Comprehensively capturing the global semantics**. Recent studies have revealed that causal LLMs tend to exhibit a bias towards capturing global information from the input samples, often resulting in a tendency to overlook information located in the middle, especially for long sequences [15, 57]. As illustrated in Figure 1(a), we chose 500 samples from WebQA [10], where the task is to find and reason about the right image-text pair among five distractors to produce a yes or no answer. We conducted experiments using VILA [52], a MLLM with state-of-the-art interleaved image-text comprehension ability, alongside our model. When placing the relevant pair in different positions, the performance of MLLMs followed a 'U' shape, indicating a bias in capturing global semantic information. Consequently, MLLMs encounter difficulties in forming comprehensive representations that encompass global semantics for retrieval tasks.

*(ii)* **Keenly differentiating the detailed semantics**. Some research [47, 82] has found that the existing generative training framework cannot fully distinguish input semantics in certain contexts, causing MLLMs to struggle with tasks requiring fine-grained semantics [46, 98]. As depicted in Figure 1(b), we noticed that MLLMs face challenges in choosing the right description for two similar images in the MMVP-VLM benchmark [81]. This indicates that MLLMs struggle to effectively differentiate the detailed semantics of input samples, naturally leading to difficulties in forming effective queries for retrieval.

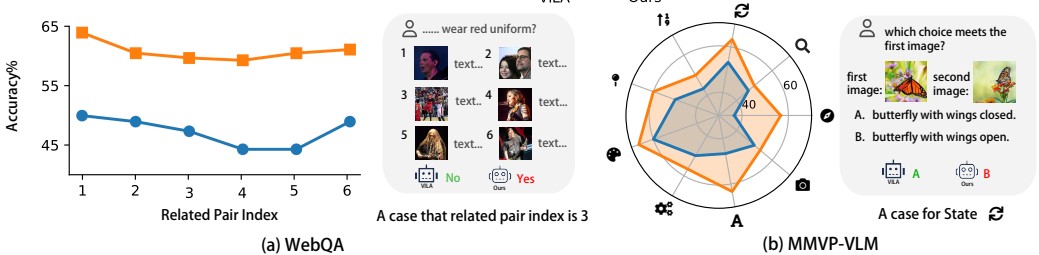

Figure 1: (a) In WebQA [10], the accuracy roughly forms a "U" shape curve when the relevant image-text pair for a question appears at different positions. While our model also shows similar trends, it tends to be more stable overall. (b) The accuracy of various types of questions in MMVP-VLM [81], it can be observed that our model's performance improves on such tasks after introducing the discriminative training. Details can be seen in Appendix E.3

In this paper, we argue that the current separated paradigms possess the potential for achieving synergistic gains. We propose **Sugar**: **S**tructure-induced approach to **u**nify **g**enerative **a**nd disc**r**iminative paradigms (shown in Figure 2), leveraging discriminative training to acquire the two abilities above while harnessing the potential of generative training in complex discriminative tasks like image-text interleaved retrieval and fine-grained retrieval. Specifically, we explicitly impose the semantic relationships between different input samples as an induced structural constraint on the hidden state of MLLMs. We consider the interleaved image-text sequence as the general format of input samples, and then formulate the relationship between any two samples as a dynamic sequence alignment problem within the Dynamic Time Warping framework [67, 33]. In this way, we can explicitly modulate the hidden states of the MLLM by leveraging the semantic relationships between interleaved input sequences, thereby encouraging the MLLM to fully **capture the global semantics** of the inputs.

To further enhance the ability to **differentiate fine-grained semantics**, we integrate a novel kernel into the Dynamic Time Warping framework. Leveraging the strengths of various discriminative pre-trained models, it performs dynamic sequence alignment for diverse embeddings tailored to specific contexts, thus addressing the inherent limitations in fully utilizing input semantics. Through this explicit structure-induced constraint, our framework enables MLLMs to capture the global semantics and fine-grained details of the input multimodal sequence more effectively, thus bridging the gap between generative and discriminative training paradigms.

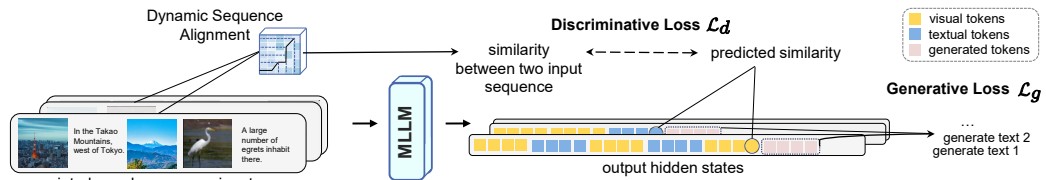

Figure 2: Our structure-induced generative and discriminative training joint training strategy.

Our method effectively balances both discriminative and generative tasks, demonstrating synergistic benefits. *(i)* Large-scale generative pre-trained models possess semantic-rich hidden states [41, 91, 23], which facilitate **discriminative tasks** like retrieval. Moreover, harnessing the capabilities of MLLM is crucial for complex discriminative tasks, such as interleaved image-text retrieval and fine-grained retrieval. *(ii)* By integrating discriminative tasks, the model's effectiveness in **generative tasks**, particularly within tasks requiring cognitive and discrimination abilities, is enhanced, thereby mitigating certain occurrences of hallucinations. *(iii)* We can employ Sugar to realize **retrieval-augmented generation** [2], eliminating the need for an off-the-shelf retrieval module [75], thereby amplifying the performance of various generative tasks. The usage of off-the-shelf retrieval presents a challenge wherein the retriever's performance affects the generator's final output [62]. This necessitates independent optimization of both components, posing a dilemma in selecting optimal configurations. However, our approach circumvents such optimization challenges.

Through extensive experimentation, we have demonstrated the effectiveness of our approach. For generative tasks, Sugar establishes new state-of-the-art results on the tasks for complicated multi-modal comprehension tasks (*i.e.,* DEMON [47]), fine-grained semantic distinctions (*i.e.,* VizWiz [28], MME [95]), object hallucinations detection (*i.e.,* POPE [51]) (Section 4.2 and Section 4.3). For discriminative tasks, we achieved competitive results in image-text retrieval compared, and significantly surpassed CLIP in interleaved retrieval and fine-grained retrieval (Section 4.4). Furthermore, employing the retrieval-augmented generation (RAG) strategy led to further improvements in a series of generative tasks (Section 4.5).

## 2 Related Work

**Multi-modal Large Language Models**. Flamingo [3] and BLIP-2 [49] integrate LLMs with visual encoders, showcasing impressive zero-shot capabilities by aligning visual features with language representations. Building upon the advancements of LLaVA-1.5 [55], subsequent studies [103, 19, 94, 6, 42, 72, 95, 98, 45] propose fine-tuning MLLMs with multimodal instruction tuning data [102]. Recently, there has been a surge in research [52, 80, 22, 21, 48] dedicated to enhancing the capacity of MLLMs to process interleaved image-text inputs effectively. However, these models primarily focus on generative tasks, overlooking the importance of introducing discriminative constraints. In this paper, we propose a structure-induced joint training strategy for unifying generative and discriminative tasks, further enhancing the capabilities of MLLMs, especially those requiring cognitive and discriminative abilities.

**Vision-Language Pre-training**. Vision-Language Pre-training primarily come in two forms: single-stream and dual-stream. In single-stream models, the embeddings for the image and text modalities are concatenated and jointly encoded [39, 50], while in dual-stream models, they are encoded by separate modality-specific encoders with optional cross-modality fusion [73, 31, 5]. These models have shown effectiveness in tasks such as classification and retrieval. However, they face challenges including difficulty in processing complex composed sequences [53, 54] and limited ability to discern detailed semantic differences [81, 79]. Recent attempts to utilize generative MLLMs for discriminative tasks have faced limitations, such as singular discriminative tasks [89], weak discriminative task performance [40], poor generalization [59], and compromised generative capabilities [8].

**LLMs for Retrieval**. Early models for retrieval primarily focused on word representations [16, 64, 74], with minimal generative capabilities. Some recent works have endeavored to fine-tune generative pre-trained LLMs to generate discriminative embeddings, albeit at the expense of compromising the model's original generative capabilities [44, 70, 65, 63, 24, 71]. GRIT [66] integrates generative and discriminative tasks in NLP and demonstrates mutual benefits between them. However, its training cost is prohibitively high compared to individual tasks. Moreover, due to its specialized attention mechanism, the model can only be trained from scratch.

**Retrieval-Augmented Generation**. Retrieval-Augmented Generation (RAG)[25, 69], which harnesses the advanced inference capabilities of LLMs along with external knowledge, has the potential to significantly mitigate issues related to long-tail entities and reduce the occurrence of hallucina-

tory responses [29, 36, 101, 77, 90, 92, 97]. Recently, there have also been related studies in the multimodal domain attempting to utilize retrieval augmentation [93, 96]. These methods typically require an additional retrieval module (*e.g.,* CLIP), leading to component optimization challenges where the overall model performance is affected by the performance of the retrieval model, as well as concerns regarding the compatibility between the retrieval model and the MLLMs. Furthermore, retrieval modules like CLIP struggle to handle compositional or fine-grained scenarios, posing certain challenges for retrieval.

## 3 Method

As illustrated in Figure 3, we initially introduce the problem formulation and offer an overview of our structure-induced joint training strategy in Section 3.1. Subsequently, we delve into the specifics of dynamic sequence alignment algorithm in Section 3.2. Finally, we further introduce the Triple Kernel to aid in discriminating detailed semantics in Section 3.3.

### 3.1 Problem Formulation and Architecture Overview

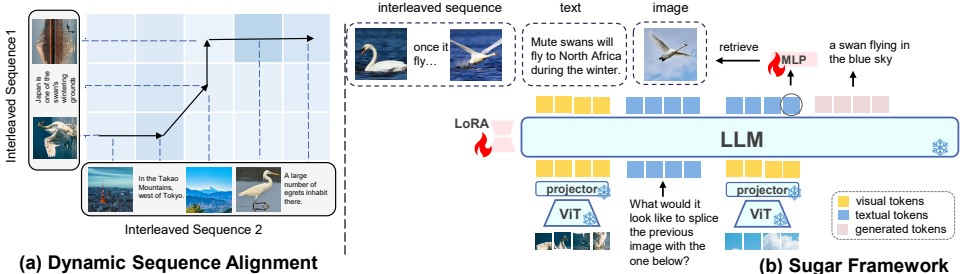

(a) **Dynamic Sequence Alignment**          (b) **Sugar Framework**

Figure 3: (a) **Dynamic Sequence Alignment**. Semantically matched slices are connected with a blue dashed line. The arrows indicate the direction of the ordered temporal alignment path. With these alignments, we can obtain the similarity between two interleaved inputs for training. (b) **Sugar Framework**. Sugar supports both multi-modal generation and retrieval simultaneously.

We view the interleaved image-text sequence as the general format for input samples, where images and textual data are alternately arranged. Typically, Multimodal Large Language Models (MLLMs) [55, 52, 11, 6] are tailored to generate text based on such input sequences, and it is conventionally optimized using self-regressive loss $\mathcal{L}_g$. A special scenario arises when the input comprises only one image and a question, prompting the MLLMs to generate an answer accordingly.

While intuitive, this optimization objective solely supervises text generation and lacks constraints on the hidden states of the entire interleaved sequence input. Additionally, the existing generative training framework struggles to fully distinguish input semantics in certain contexts, such as discerning fine-grained object details. Consequently, it fails to adequately capture the global information or distinguish detailed semantics of the input samples.

Hence, we introduce a structure-induced constraint $\mathcal{L}_d$ (see in Figure 2), which explicitly imposes the semantic relationships between different input samples as an induced structural constraint on the hidden states of MLLMs, facilitating the model in **capturing global semantics**. We conceptualize the derivation of semantic relationships between input samples as a Dynamic Sequence Alignment problem [67]. Additionally, we straightforwardly select a token in the hidden state of the MLLM to encompass all preceding input information, eliminating the need for training any specialized tokens.

To further effectively **distinguish detailed semantics**, we integrate a novel kernel into the Dynamic Time Warping framework. Leveraging the strengths of various discriminative pre-trained models. Combined with this newly proposed loss with a hyperparameter $\alpha$, the training objective can be formulated as:

$$\mathcal{L} = \mathcal{L}_g + \alpha \mathcal{L}_d \tag{1}$$

### 3.2 Dynamic Sequence Alignment

We formulate the computation of relationships within input interleaved sequences as a dynamic sequence alignment problem, and solve it by global alignment kernel. For two interleaved image-text sequence, each consisting of $n$ and $m$ images/sentences in total respectively (which we'll refer to as slices later on). We encode and normalize each slice, resulting in two sequences $\mathbf{x} = (x_1, \ldots, x_n)$ and

$\mathbf{y} = (y_1, \ldots, y_m)$ all of which take values in a state space $\mathcal{X}$, that is two elements of $\mathcal{X}^\star \overset{\text{def}}{=} \bigcup_{i=1}^\infty \mathcal{X}^i$. In our setting, $\mathcal{X}$ is simply $\mathbb{R}^d$, $d$ refers to the feature dimension. We define the global alignment kernel as follows, and it has been proved to be positive-definite under mild conditions and may prove more robust to quantify the similarity of two sequences [73, 68]:

$$K(\mathbf{x}, \mathbf{y}) = \sum_{\pi \in \mathcal{A}(\mathbf{x}, \mathbf{y})} \prod_{i=1}^{|\pi|} e^{-\phi_\sigma} \in (0, 1] \tag{2}$$

Following the suggestion by [18], we let $\varphi_\sigma = \frac{1}{2\sigma^2} \varphi\left(x_{\pi_1(i)}, y_{\pi_2(i)}\right) + \log(2 - e^{-\frac{\varphi\left(x_{\pi_1(i)}, y_{\pi_2(i)}\right)}{2\sigma^2}})$, $\sigma$ is standard deviation, and it can be calculated by $\sigma = \delta\sqrt{\frac{M+N}{2}}$ for $x_i, y_i$ in $\mathbf{x}, \mathbf{y}$. $\delta$ is a fixed pre-defined hyperparameter and $\varphi\left(x_{\pi_1(i)}, y_{\pi_2(i)}\right)$ is the distance between slice $x_{\pi_1(i)}$ and $y_{\pi_2(i)}$ for an alignment (details for the definition of alignment can be seen in Appendix D.2).

Due to the causal attention mechanism, the token in hidden state of MLLM can encapsulate information from preceding tokens in the sequence. Therefore, we directly utilize the last token $d_i$ of a sequence from the MLLM's hidden state and map it to the $r_i$ using an MLP to represent the entire in-context sequence. During training, we obtain a set of $(r_1, r_2, \ldots, r_n)$ and their corresponding input sequence embedding set $(\mathbf{x}_1, \mathbf{x}_2, \ldots, \mathbf{x}_n)$. It is noteworthy that $r_i$ and $r_j$ ($\mathbf{x}_i$ and $\mathbf{x}_j$) may originate from the same sequence but occupy different positions, thus enabling our method to utilize samples more efficiently.

Leveraging the GAK, we can derive the similarity matrix of $(r_1, r_2, \ldots, r_n)$ and $(\mathbf{x}_1, \mathbf{x}_2, \ldots, \mathbf{x}_n)$ distinctively, denoted as $\mathcal{M}^r, \mathcal{M}^l \in \mathbb{R}^{n \times n}$. For imposing the semantic relationships between different input samples as an induced structural constraint on the hidden state of MLLMs, we employ Mean Squared Error (MSE) loss aligned $\mathcal{M}^r$ with the label matrix $\mathcal{M}^l$. This approach eliminates the need for pre-defined label (*i.e.,* positive and negative candidates) during training, allowing seamless integration into the aforementioned training framework (for specific training templates, please refer to Appendix E.1). Thus, we have the discriminative loss $\mathcal{L}_d$:

$$\mathcal{L}_d = \frac{1}{n} \sum_{i=1}^n \sum_{j=1}^n \left(m_{ij}^r - m_{ij}^l\right)^2 \tag{3}$$

Additionally, when both $\mathbf{x}$ and $\mathbf{y}$ contain only one slice, the computed result of the formula is monotonically increasing with the directly calculated cosine similarity (proof can be seen in Appendix 3). Therefore, in such cases, we simplify the computation by directly using cosine similarity. If $r_i$ and $r_j$ comes from the same input interleaved sample, we manually set $m_{ij}^l = 1$.

### 3.3 Detailed Semantics Modeling

To further effectively distinguish detailed semantics, we further propose the Triple Kernel (TK), a positive definite kernel compatible with the previous framework. The TK leverages representations from diverse pre-trained discriminative models across uni-modal and cross-modal settings, harnessing their respective strengths. The definition is as below:

For two slice $a, b \in \mathbb{R}^d$, meets (i) $|a| = |b| = 2$, $d = d_1 + d_2$, $a = \text{concat}(a_1, a_2), b = \text{concat}(b_1, b_2)$, $a_1, b_1 \in \mathbb{R}^{d_1}, a_2, b_2 \in \mathbb{R}^{d_2}$ and $|a_1| = |a_2| = |b_1| = |b_2| = 1$; or (ii) $|a| = |b| = 1$. We define tripe kernel as follows:

$$\varphi(a, b) = \begin{cases} \|a_1 - b_1\|^2 & |a| = |b| = 2 \text{ and } a, b \text{ in uni-modal} \\ \|a_2 - b_2\|^2 & |a| = |b| = 2 \text{ and } a, b \text{ in cross-modal} \\ \|a - b\|^2 & \text{else} \end{cases} \tag{4}$$

We prove triple kernel $\varphi$ is a conditionally positive-definite kernel defined on $\mathcal{X} \times \mathcal{X} \to \mathbb{R}$ (Appendix 2), aligning with the kernel definition in [18], thereby possessing its properties.

In practice, we let the feature dimension $d = d_1 + d_2$. For images, we employ DINOv2-base [68] and CLIP ViT-L/14 [73] for encoding, then concatenate the embeddings after normalization. For sentences, we utilize BGE-base [87] and CLIP ViT-L/14, keeping the dimension unchanged. By utilizing the Triple Kernel, we can fully leverage the strengths of these three models, effectively distinguishing detailed semantics.

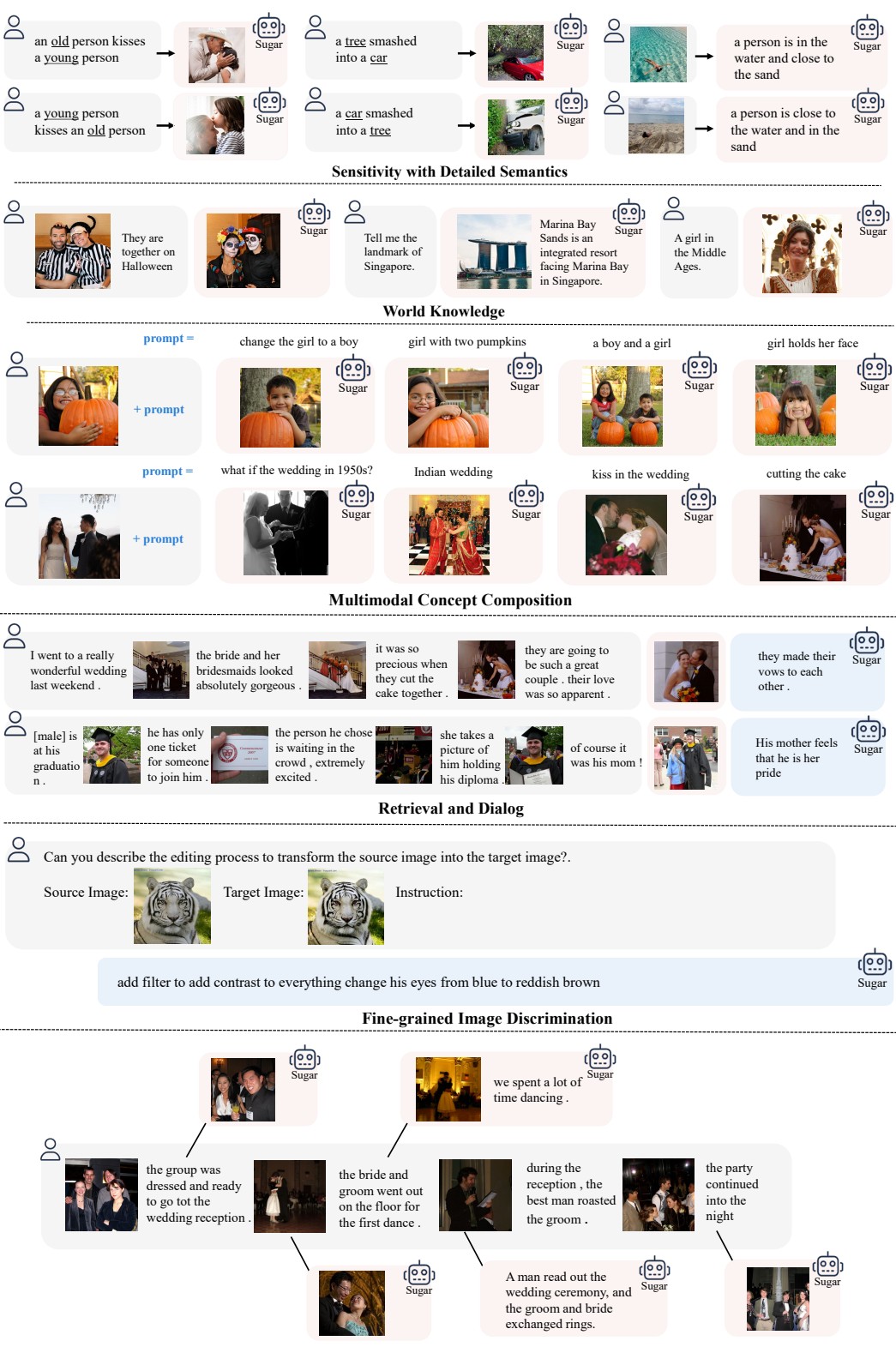

Figure 4: Selected examples for various image-text tasks. The pink background indicates retrieval results, while the blue background indicates generated results. More examples are provided in the Appendix F.2.

| Method | LLM | Res. | VQA$^{v2}$ | GQA | VizWiz | SQA$^I$ | VQA$^T$ | POPE | MME$^P$ | MME$^C$ | MMB | LLaVA$^{Wd}$ | MM-Vet |
|--------|-----|------|------|-----|--------|------|------|------|------|------|-----|--------|--------|
| BLIP-2 [49] | Vicuna-13B | 224 | 41.0 | 41 | 19.6 | 61 | 42.5 | 85.3 | 1293.8 | 290.0 | – | 29.1 | 22.4 |
| InstructBLIP [19] | Vicuna-7B | 224 | – | 49.2 | 34.5 | 60.5 | 50.1 | – | – | – | 36 | – | 26.2 |
| InstructBLIP [19] | Vicuna-13B | 224 | – | 49.5 | 33.4 | 63.1 | 50.7 | 78.9 | 1212.8 | 291.9 | – | – | 25.6 |
| Shikra [12] | Vicuna-13B | 224 | 77.4* | – | – | – | – | – | – | – | 58.8 | – | – |
| IDEFICS-9B [42] | LLaMA-7B | 224 | 50.9 | 38.4 | 35.5 | – | 25.9 | – | – | – | 48.2 | – | – |
| IDEFICS-80B [42] | LLaMA-65B | 224 | 60.0 | 45.2 | 36.0 | – | 30.9 | – | – | – | 54.5 | – | – |
| Qwen-VL [6] | Qwen-7B | 448 | 78.8* | 59.3* | 35.2 | 67.1 | 63.8 | – | – | – | 38.2 | – | – |
| Qwen-VL-Chat [6] | Qwen-7B | 448 | 78.2* | 57.5* | 38.9 | 68.2 | 61.5 | – | 1487.5 | **360.7** | 60.6 | – | – |
| LLaVA-1.5 [55] | Vicuna-7B | 336 | 78.5* | 62.0* | 50.0 | 66.8 | 58.2 | 85.9 | 1510.7 | – | 64.3 | 49.0 | 30.5 |
| VILA-7B [52] | Llama-2-7B | 336 | 79.9* | 62.3* | 57.8 | 68.2 | 64.4 | 85.5 | 1533.0 | 296.1 | **68.9** | 70.0 | **34.9** |
| **Sugar** | Vicuna-7B | 336 | 76.0* | 58.7* | **60.4** | 69.4 | 57.5 | **86.6** | 1550.8 | 300.0 | 64.9 | **75.6** | 31.3 |

Table 1: Comparison with state-of-the-art methods on 11 visual-language benchmarks. We mark the best performance **bold** and the second-best underlined. Benchmark names are abbreviated due to space limits. VQA-v2 [27]; GQA [35]; VizWiz [28]; SQA$^I$: ScienceQA-IMG [61]; VQA$^T$: TextVQA [76]; POPE [51]; MME$^P$, MME$^C$: MME Perception, MME Cognition [95]; MMB: MMBench [58]; LLaVA$^{Wd}$: LLaVA-Bench(In-the-Wild)-Detail [56]; MM-Vet [99]. * indicates the training images of the datasets are observed during training.

## 4 Experiments

To assess Sugar's **generative** ability, we conduct a comprehensive comparison with state-of-the-art models on 11 commonly used visual-language benchmarks in Section 4.2. Furthermore, we evaluate more complicated multimodal comprehension tasks on DEMON with 29 datasets in Section 4.3. For **discriminative** tasks, we compare performance across three different retrieval tasks: image-text retrieval, interleaved retrieval, and fine-grained retrieval in Section 4.4. Subsequently, we leverage Sugar's discriminative ability for **retrieval-augmented generation** compared with common used retrieve module in Section 4.5. Finally, we conduct ablation experiments to analyze the effectiveness of our method in Section 4.6.

### 4.1 Setup

We apply our method to VILA [52], a recent state-of-the-art MLLM supporting interleaved input. We further fine-tune VILA using LoRA [30]. Details about the experiments setting, datasets and the instruction examples, please check in Appendix E.

### 4.2 Multimodal Comprehension on 11 Benchmarks

We conduct a comprehensive comparison with state-of-the-art models on 11 commonly used benchmarks, as shown in Table 1. Compared to existing models, Sugar achieves remarkable improvements over the second-best performing model on tasks requiring fine-grained semantics (*i.e.,* LLaVA$^{Wd}$ [56], VizWiz [28], SQA [61] improve by 8%, 4.5%, 1.8% respectively) and benchmarks for detecting hallucinations (*i.e.,* POPE [51]), while maintaining competitive results in other tasks. Notably, Sugar excels in discriminative tasks and still achieves 5 state-of-the-art results and 3 second-best results on 11 benchmarks for generative tasks, even outperforming some models larger than 7B. Our results demonstrate the benefits of incorporating the discriminative loss, aiding in fine-grained semantic tasks and reducing hallucinations.

### 4.3 Complicated Multimodal Comprehension on DEMON

Table 2 demonstrates the superior performance of Sugar on the DEMON benchmark, which comprises 7 categories and a total of 29 sub-tasks. These tasks are considerably more complex than the previously used 11 common benchmarks. DEMON is tailored to evaluate the capacity of models and systems to understand demonstrative instructions that include multiple, interleaved, and multimodal contexts, presenting the essential information needed to complete a task. Sugar surpasses the previous state-of-the-art model on the DEMON benchmark, VPG-C [47], across 6 of 7 categories. For example, we achieve performance improvements of 36.1% in Text-Rich Images QA (TRQA) tasks and 17.2% in Visual Relation Inference (VRI) tasks, both of which require detailed semantics, compared to the second-best performing model. This underscores our advanced ability to associate interleaved

| | LLM | MMD | VST | VRI | MMC | KGQA | TRQA | MMR |
|---|---|---|---|---|---|---|---|---|
| OpenFlamingo [4] | MPT-7B | 16.9 | 24.2 | 13.9 | 21.7 | 32.0 | 30.6 | 41.6 |
| BLIP-2 [49] | Vicuna-13B | 26.1 | 21.3 | 10.7 | 17.9 | 39.2 | 33.5 | 39.7 |
| InstructBLIP [19] | Vicuna-7B | 33.6 | 24.4 | 11.5 | 21.2 | 47.4 | 44.4 | 48.6 |
| MiniGPT-4 [103] | Vicuna-7B | 13.7 | 17.1 | 8.0 | 16.6 | 30.3 | 26.4 | 43.5 |
| LLaVA [56] | Vicuna-7B | 7.8 | 10.7 | 8.3 | 15.9 | 36.2 | 28.3 | 41.5 |
| mPlug-Owl [94] | LLaMA-7B | 12.7 | 19.3 | 5.4 | 16.3 | 33.3 | 32.5 | 42.5 |
| VPG-C [47] | Vicuna-7B | 37.5 | 25.2 | 25.9 | **22.2** | 48.6 | 44.9 | 50.3 |
| VILA-7B [52] | Vicuna-7B | 47.8 | 25.8 | 13.2 | 17.2 | 60.1 | 42.1 | 50.5 |
| **Sugar** | Vicuna-7B | **51.8** | **34.3** | **32.3** | 16.8 | **64.4** | **65.9** | **51.7** |

Table 2: Comparision with state-of-the-art method on DEMON [47] benchmark.

**(a) MSCOCO**

| Model | R@1 | R@5 | R@10 |
|---|---|---|---|
| Text → Image | | | |
| FROMAGe(d) | **23.4** | 47.3 | 59.0 |
| FROMAGe(g+d) | 23.4 | 47.2 | 58.0 |
| **Sugar** | 22.0 | **49.1** | **63.1** |
| Image → Text | | | |
| FROMAGe(d) | **26.8** | 52.4 | 63.6 |
| FROMAGe(g+d) | 26.4 | 52.3 | 63.4 |
| **Sugar** | 25.6 | **53.6** | **66.7** |

**(b) VIST**

| Model | Inputs | R@1 | R@5 | R@10 |
|---|---|---|---|---|
| CLIP ViT-L/14 | 5c | 5.9 | 19.5 | 28.0 |
| FROMAGe | 5c | **11.9** | 23.8 | 31.7 |
| **Sugar** | 5c | 10.1 | **26.3** | **36.2** |
| BLIP[†] | 5c | 6.2 | 16.8 | 23.4 |
| CLIP ViT-L/14[†] | 5c | 8.8 | 22.3 | 29.8 |
| FROMAGe[†] | 5c | **13.2** | 28.5 | 36.7 |
| **Sugar[†]** | 5c | 11.0 | 27.3 | **37.0** |
| CLIP ViT-L/14 | 5c+4i | 2.4 | 21.3 | 34.0 |
| FROMAGe[†] | 5c+4i | 18.2 | 42.7 | 51.8 |
| **Sugar[†]** | 5c+4i | **21.9** | **46.7** | **59.2** |

**(c) Winoground**

| Model | Text | Image | Group |
|---|---|---|---|
| VinVL | 37.8 | 17.8 | 14.5 |
| UNITER$_{large}$ | 38.0 | 14.0 | 10.5 |
| VisualBERT$_{base}$ | 15.5 | 2.5 | 1.5 |
| ViLLA$_{large}$ | 37.0 | 13.25 | 10.0 |
| ViLT ViT-B/32 | 34.8 | 14.0 | 9.3 |
| LXMERT | 19.3 | 7.0 | 4.0 |
| ViLBERT$_{base}$ | 23.8 | 7.3 | 4.8 |
| FLAVA$_{ITM}$ | 32.3 | 20.5 | 14.3 |
| FLAVA$_{contrastive}$ | 25.3 | 13.5 | 9.0 |
| CLIP ViT-B/32 | 30.8 | 10.5 | 8.0 |
| **Sugar** | **40.0** | **36.3** | **27.0** |

Table 3: Retrieval results compared with previous models, reported by Recall@$k$ for (a)(b) and Accuracy (%) for (c). **(a) MSCOCO** for image-text retrieval: FROMAGe(d) indicates the FROMAGe model pre-trained only with discriminative loss, and FROMAGe(g+d) indicates joint training with both discriminative and generative losses. **(b) VIST** for interleaved retrieval: [†] indicates retrieval over images not previously seen in the story sequence. "5c+4i" is shorthand for 5 captions and 4 images, and "5c" is shorthand for 5 captions. **(c) Winoground** for fine-grained retrieval.

text-image inputs for stronger in-context understanding, and Sugar's strong capability to capture global semantics in interleaved sequences, facilitated by joint training with discriminative loss.

### 4.4 Zero-shot Cross-modal Information Retrieval

**Image-text Retrieval.** We evaluated the performance of Sugar on the widely adopted MSCOCO [38] dataset in the context of a standard image-text retrieval task. Sugar demonstrated comparable performance to FROMAGe [40] in R@1 and surpassed it in R@5 and R@10, highlighting Sugar's superiority in normal retrieval tasks.

**Interleaved Retrieval.** To assess the proficiency of Sugar in processing multimodal contextual information, we evaluated its performance in retrieving relevant images conditioned on sequences of interleaved image-text inputs from the Visual Storytelling (VIST) dataset [32]. We conducted evaluations across several experimental configurations, following the same setup as FROMAGe [40] (see Appendix F.1). Our results show that Sugar outperforms FROMAGe in most settings, particularly achieving a 20.3% improvement in the 5c+4i configuration, significantly surpassing both CLIP and BLIP-2. This demonstrates that our method effectively leverages MLLMs' ability to handle complex interleaved sequence inputs, thereby achieving superior retrieval performance.

**Fine-grained Retrieval.** We tested fine-grained retrieval using the Winoground dataset [79], which evaluates the ability to perform vision-linguistic compositional reasoning. Surprisingly, Sugar outperformed all discriminative pre-trained models (both single-stream and dual-stream encoder architectures), achieving improvements of 5.3%, 77.1%, and 86.2% over the second-best model in the Text, Image, and Group dimensions, respectively. This demonstrates Sugar's strong capability to distinguish detailed semantics and performing compositional reasoning.

### 4.5 Retrieval-Augmented Generation

Due to Sugar's dual capabilities in both discrimination and generation, we can achieve retrieval augmentation without the need for an additional retrieval module. For performing retrieval-augmented generation (RAG), we selected two tasks, namely VizWiz and SQA$^I$, as they offer held-in data that were not seen during model training. We utilized a mixed set comprising the widely-used LLaVA-1.5 SFT subset and the held-in datasets of the two tasks as the knowledge base and employed different

|  | (a) Retrieval-Augmented Generation | |
|---|---|---|
|  | VizWiz | SQA[I] |
| LLaVA-1.5 | 50.0 | 66.8 |
| +CLIP | 42.6 (−14.8%) | 62.0 (−8.6%) |
| +BLIP2 | 43.0 (−14.0%) | 62.5 (−6.4%) |
| VILA | 57.8 | 64.4 |
| +CLIP | 49.3 (−14.7%) | 65.7 (+0.6%) |
| +BLIP2 | 49.6 (−14.2%) | 66.1 (+2.6%) |
| **Sugar** | 60.4 | 69.4 |
| +RAG | **61.9** (+2.5%) | **71.9** (+3.6%) |

**(b) Ablation Study**

|  | Generative Tasks | | | Discriminative Tasks | | |
|---|---|---|---|---|---|---|
|  | SQA[I] | POPE | KGQA | MSCOCO | VIST | Winoground |
| **Sugar** | 72.6 | **86.6** | 64.4 | 49.1 | **46.7** | **36.3** |
| w/o data$_d$ | **72.8** | 85.3 | 61.7 | **49.6** | 42.0 | 34.8 |
| w/o data$_g$ | 68.0 | 86.4 | 62.5 | 46.0 | 40.7 | 33.5 |
| w/o GAK | 72.1 | 86.0 | 61.1 | 48.2 | 33.5 | 29.3 |
| w/o TK | 71.6 | 85.3 | 63.9 | 49.0 | 44.1 | 20.5 |
| w/o AvgPool | 72.4 | 86.2 | **64.6** | 39.7 | 38.1 | 31.5 |

Table 4: (a) **Retrieval-Augmented Generation.** (b) **Ablation Study.** For MSCOCO, we report the $R@5$ in text-to-image retrieval. For VIST, we report the $R@5$ of retrieving an image given 5 captions and 4 images. For Winoground, we report the Image score. For other tasks, we report Accuracy (%).

retrieval modules to retrieve relevant knowledge for the MLLM. The results are as follows: *(i)* We observed a drop in performance for LLaVA-1.5 with RAG in all tasks. This may be because LLaVA is designed solely for single-image input, without the ability to utilize in-context external knowledge. *(ii)* Compared to VILA, Sugar's performance improved in both tasks, whereas VILA improved in SQA[I] but decreased in VizWiz. These findings suggest that Sugar's retrieved knowledge is more beneficial, while the knowledge retrieved by CLIP and BLIP-2 may hinder performance.

## 4.6 Ablations

**Importance of Both Tasks. (1) w/o data$_g$**: As shown in Table 4, when we reduce the amount of data for discriminative tasks (Row 2), there are performance drops of 1.5% in hallucination detection tasks (*i.e.,* POPE) and 4.2% in interleaved multi-modal comprehension tasks (*i.e.,* KGQA). **(2) w/o data$_d$**: Similarly, reducing the data for generative tasks (Row 3), the performance on generative tasks declines with a 10.1% decrease in VIST, which requires global semantics capturing, and a 4.1% decrease in Winoground, which necessitates fine-grained semantic understanding. This indicates that **generative and discriminative training can mutually benefit each other**.

**Effectiveness of Individual Components. (1) w/o GAK**: When we exclude the Global Alignment Kernel (GAK) (Row 4) and resort to using the average similarity for the slices, a notable decrease in performance is observed across several interleaved image-text tasks (*i.e.,* a 5.1% decrease in KQGA and a 28.3% decrease in VIST). This underscores the fundamental role of GAK in aiding Sugar to capture global semantics effectively. **(2) w/o TK**: Upon removal of the Triple Kernel (TK) (Row 5) and utilization of CLIP for encoding the input sequence instead, a dramatic performance decline is evident in Winoground, with a 43.5% decrease. This underscores the significant role of TK in facilitating the distinction of detailed semantics. **(3) w/o AvgPool**: When solely using the last token for retrieval, a general decline in performance is observed across discriminative tasks, with decreases of 19.1% for MSCOCO, 22.6% for VIST, and 13.2% for Winoground. This phenomenon may be attributed to the last token of an image often corresponding to a pooling token, containing relatively weaker semantic information. Utilizing all image tokens and performing AvgPooling tends to yield greater improvements in retrieval tasks.

## 5 Conclusion

Vision-Language Models (VLMs) have been trained using both generative and discriminative paradigms, each with distinct advantages and limitations. To bridge this gap, we introduce **S**tructure-induced approach to **u**nify **g**enerative **a**nd disc**r**iminative paradigms, which imposes semantic relationships between input samples, thereby enhancing the MLLM's ability to capture global semantics and distinguish fine-grained details. This approach effectively balances generative and discriminative tasks, yielding synergistic benefits. Extensive experiments demonstrate the effectiveness of our approach, achieving state-of-the-art results in multiple generative tasks, particularly those requiring cognitive and discrimination abilities, while also demonstrating competitive performance in discriminative tasks such as image-text retrieval and achieving state-of-the-art results in interleaved and fine-grained retrieval. Furthermore, employing a retrieval-augmented generation strategy within a single model leads to additional improvements, offering a promising direction for future research.

**Acknowledgment.** This work has been supported in part by the Key Research and Development Projects in Zhejiang Province (No. 2024C01106, 2024C01028), the NSFC (No. 62272411), the National Key Research and Development Project of China (2018AAA0101900), and Ant Group.

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

# A    Broader Impact

The broader impact of Sugar, carries both potential benefits and risks upon deployment and release. Some considerations are unique due to their visual nature, while others mirror existing instruction-following Large Language Models (LLMs). Built upon Vicuna, CLIP, DINOv2, and BGE, Sugar inherits issues associated with LLMs and vision encoders. Below, we outline risks and mitigation strategies for its release.

**Hallucination.**    Similar to other MLLMs, Sugar may generate outputs detached from facts or input data, posing concerns, especially in critical applications like medicine and the field related to security.

**Biases.**    Bias from base models can transfer to Sugar, originating from both the vision encoder (CLIP) and the language decoder (Vicuna), potentially leading to biased outcomes or unfair representations.

**Ethical Impacts.**    This study doesn't raise ethical concerns, as it doesn't involve subjective assessments or private data, only utilizing publicly available datasets.

**Expected Societal Implications.**    A significant societal concern lies in potential misuse, such as fabricating unauthorized texts leading to misinformation, privacy breaches, and other damaging consequences. Strong ethical standards and ongoing surveillance are essential for mitigation.

These issues aren't unique to our method but are prevalent across different techniques for multi-concept customization. Despite the risks, we believe the benefits outweigh the potential harm, allowing ongoing investigation and improvement of the model while engaging the community in developing better mitigation strategies. Moreover, its release can foster new applications and research directions, contributing to the progress and responsible deployment of foundation models in vision-language tasks.

# B    Limitations

*(i)* Our method, while effective, may inherit limitations from the underlying models, such as hallucination in generating outputs detached from facts or input data and potential biases originating from the model we used. *(ii)* The training data might inevitably contain mismatched image and text, which could adversely affect training.

# C    Mathematical Proof

**Theorem 1.** *The alignment kernel $K$ can be computed in quadratic complexity, namely in $O(mnd^2)$ iterations. where $m, n$ denotes the length of two sequence and their hidden dimension all is $d$, $m, n, d \in \mathbb{R}$.*

*Proof.* Given $\mathbf{x} = (x_1, \ldots, x_n)$ and $\mathbf{y} = (y_1, \ldots, y_m)$ two sequences of $\mathcal{X}^\star$, we set the double-subscripted series $M_{i,j}$ as $M_{i,0} = 0$ for $i = 1, ..., n$, $M_{0,j} = 0$ for $j = 1, ..., m$, and $M_{0,0} = 1$. Computing recursively for $(i, j) \in \{1, ..., n\} \times \{1, ..., m\}$ the terms

$$M_{i,j} = (M_{i,j-1} + M_{i-1,j-1} + M_{i-1,j})k(x_i, y_j)$$

we obtain that $K(\mathbf{x}, \mathbf{y}) = M_{n,m}$ The result can be proved by recursion and is intuitively an equivalent of the Dynamic Time Warping(DTW) [67, 17] algorithm where the max-sum algebra is simply replaced by the sum-product one [18]. □

**Theorem 2.** *triple kernel $\varphi$ is a conditionally symmetric positive-definite kernel [18] defined on $\mathcal{X} \times \mathcal{X} \to \mathbb{R}$.*

*Proof.* (i) for two slice $a, b \in \mathbb{R}^d$, meets $|a| = |b| = 2$ , $d = d_1 + d_2$, $a = \text{concat}(a_1, a_2), b = \text{concat}(b_1, b_2)$, $a_1, b_1 \in \mathbb{R}^{d_1}, a_2, b_2 \in \mathbb{R}^{d_2}$ and $|a_1| = |a_2| = |b_1| = |b_2| = 1$:

let $a' = \text{concat}(a'_1, a'_2, a'_3)$, $a' = \text{concat}(b'_1, b'_2, b'_3)$: when $a'(b')$ is from text modal, we let $a' = \text{concat}(a_1, a_2, \mathbf{0})(b' = \text{concat}(b_1, b_2, \mathbf{0}))$, and $a' = \text{concat}(a_1, \mathbf{0}, a_2)(b' = \text{concat}(b_1, \mathbf{0}, b_2))$ for image modal. $\mathbf{0} \in \mathbb{R}^{d_2}$. we can unify the Equation 4 first and second case in:

$$
\begin{aligned}
\varphi(a,b) &= \varphi(a', b') \\
&= (|a'_2||b'_2| + |a'_3||b'_3|)||a'_1 - b'_1||^2 + (1 - |a'_2||b'_2|)||a'_2 - b'_2||^2 \\
&\quad + (1 - |a'_3||b'_3|)||a'_3 - b'_3||^2 \\
&\geq (|a'_2||b'_2| + |a'_3||b'_3|)||a'_1 - b'_1||^2 + 0 + 0 \\
&\geq 0
\end{aligned}
$$

As $(1 - |a'_2||b'_2|) \geq 0$ and $(1 - |a'_3||b'_3|) \geq 0$. for any family $\alpha_1, \alpha_2, ...\alpha_n \in \mathcal{X}$ and $c_1, c_2, ..., c_n \in \mathbb{R}$, we have that

$$
\sum_{i,j} c_i c_j \varphi(x_i, x_j) = \sum_{i,j} c_i c_j \varphi(x'_i, x'_j) \geq 0
$$

(ii) for two slice meets $|a| = |b| = 1$: for any family $\alpha_1, \alpha_2, ...\alpha_n \in \mathcal{X}$ and $c_1, c_2, ..., c_n \in \mathbb{R}$, we have that

$$
\sum_{i,j} c_i c_j \varphi(x_i, x_j) = \sum_{i,j} c_i c_j ||x_i - x_j||^2 \geq 0
$$

Additionally, it's evident that for both (i) and (ii), $\varphi(a,b) = \varphi(b,a)$. Therefore, triple kernel $\varphi$ is a conditionally symmetric positive-definite kernel defined on $\mathcal{X} \times \mathcal{X} \to \mathbb{R}$ □

**Theorem 3.** *when both $\mathbf{x}$ and $\mathbf{y}$ contain only one slice, GAK is monotonically increasing with the directly calculated cosine similarity.*

*Proof.* let $\mathbf{x} = (x), \mathbf{y} = (y)$, and cosine similarity of $x$ and $y$ is $cos = cos < x, y >$. we can get

$$
\sigma = \delta\sqrt{\frac{M+N}{2}} = \delta\sqrt{\frac{1+1}{2}} = \delta
$$

and we have $\varphi(a,b) = |a|^2 + |b|^2 - 2cos < a, b >= 2(1 - cos < a, b >)$, thus:

$$
\begin{aligned}
\varphi_\sigma &= \frac{1}{2\sigma^2} \varphi\left(x_{\pi_1(i)}, y_{\pi_2(i)}\right) + \log\left(2 - e^{-\frac{\varphi\left(x_{\pi_1(i)}, y_{\pi_2(i)}\right)}{2\sigma^2}}\right) \\
&= \frac{1}{2\delta^2} \varphi\left(x_{\pi_1(i)}, y_{\pi_2(i)}\right) + \log\left(2 - e^{-\frac{\varphi\left(x_{\pi_1(i)}, y_{\pi_2(i)}\right)}{2\delta^2}}\right) \\
&= \frac{1 - cos < a, b >}{\delta^2} + \log\left(2 - e^{-\frac{1 - cos < a, b >}{\delta^2}}\right)
\end{aligned}
$$

Letting $t = \frac{1 - cos < a, b >}{\delta^2}$ and substituting the result of $\varphi_\sigma = t + \log(2 - e^{-t})$ into Equation 2, we obtain:

$$
\begin{aligned}
K(\mathbf{x}, \mathbf{y}) &= \sum_{\pi \in \mathcal{A}(\mathbf{x}, \mathbf{y})} \prod_{i=1}^{|\pi|} e^{-\phi_\sigma} \\
&= e^{-\phi_\sigma} \\
&= e^{-t + \log\left(\frac{1}{2 - e^{-t}}\right)} \\
&= \frac{e^{-t}}{2 - e^{-t}}
\end{aligned}
$$

Letting $s = e^{-t}$, we can further obtain:

$$
K(\mathbf{x}, \mathbf{y}) = \frac{s}{2 - s}
$$

As $\cos\langle a, b\rangle \in [-1, 1]$, $s$ strictly increases with $\cos\langle a, b\rangle$ and $s \in [e^{-\frac{2}{\delta^2}}, 1]$ when the hyperparameter $\delta$ is fixed. Derivative of $K(\mathbf{x}, \mathbf{y})$ can be obtained:

$$K(\mathbf{x}, \mathbf{y})' = \frac{2 - s + s}{(2 - s)^2} = \frac{2}{(2 - s)^2} > 0$$

Overall, when both $\mathbf{x}$ and $\mathbf{y}$ contain only one slice, GAK is monotonically increasing with the directly calculated cosine similarity. □

# D   Method Details

## D.1   Architecture Details

We adopt the manifold multimodal model architecture [55, 52, 11, 6, 34], formulated as follows:

*Visual Representation.* We first process $x_{\text{img}}$ subject to a visual representation backbone $V_\omega$ that outputs a sequence of features $p_{\text{img}} \in \mathbb{R}^{L \times h_{\text{vision}}}$ where $p_{\text{img}} = V_\omega(x_{\text{img}})$. As an example, $p_{\text{img}}$ might be the patch features output by a Vision Transformer.

*Vision-Language Projector.* Next, we map $p_{\text{img}}$ to a sequence of *embeddings* $e_{\text{img}} \in \mathbb{R}^{L \times h_{\text{text}}}$ via a learned projector $F_\psi$, where $e_{\text{img}} = F_\psi(p_{\text{img}})$.

*Language Model.* Finally, we concatenate the sequence $e_{\text{img}}$ with the text prompt embeddings $e_{\text{prompt}} = \text{embed}(u_{\text{prompt}})$, passing the result to the language model. Generally, we have the interleaved image-text input $x_{\text{input}}$ by concatting all the $e_{\text{prompt}}$ and $e_{\text{img}}$. The language model generates output text $u_{\text{gen}} = \text{LM}_\theta(x_{\text{input}})$.

*Retrieval Projector.* For discriminative tasks, we select the token $d_i$ from MLLM's hidden state and map it to $r_i$ via a learned projector $F_\varphi$.

In Implementation, we utilize CLIP ViT-L/14 [73] as the visual encoder, and Vicuna 1.5 [14] as the language model.

## D.2   Sequence Alignment

An alignment $\pi$ of length $|\pi| = p$ between two sequences $\mathbf{x}$ and $\mathbf{y}$ is a pair of increasing p-tuples $(\pi_1, \pi_2)$ such that

$$\begin{aligned} 1 = \pi_1(1) \leq ... \leq \pi_1(p) = n, \\ 1 = \pi_2(1) \leq ... \leq \pi_2(p) = m, \end{aligned} \tag{5}$$

We write $\mathcal{A}(\mathbf{x}, \mathbf{y})$ for the set of all possible alignments between $\mathbf{x}$ and $\mathbf{y}$. Intuitively, an alignment $\pi$ between $\mathbf{x}$ and $\mathbf{y}$ describes a way to associate each element of a sequence $\mathbf{x}$ to one or possibly more elements in $\mathbf{y}$, and vice versa. Such alignments can be conveniently represented by paths in the $n \times m$ grid displayed in the left of Figure 3.

with unitary increments and no simultaneous repetitions, that is $\forall 1 \leq i \leq p - 1$,

$$\begin{aligned} \pi_1(i + 1) \leq \pi_1(i) + 1, \quad \pi_2(j + 1) \leq \pi_2(j) + 1, \\ (\pi_1(i + 1) - \pi_1(i)) + (\pi_2(i + 1) - \pi_2(i)) \geq 1. \end{aligned} \tag{6}$$

The score on a path is defined as:

$$S(\pi) = \sum_{i=1}^{|\pi|} \varphi(x_{\pi_1(i)}, y_{\pi_2(i)}) \tag{7}$$

# E Experimental Details

## E.1 Datasets

**Training Data.** Our vision-language task datasets are a subset of VILA [52], including MMC4 [104], COYO [9], LLaVA-1.5 SFT dataset [55].

We use a prompt template formatted as (system-message is a system prompt from Vicuna, and the following messages all have the same meaning.):

```
{system-message}. USER: <image>\n {question}. ASSISTANT: {answer}.
```

For interleaved vision-language datasets, the template is formatted as:

```
{system-message}. USER: {interleaving question}. ASSISTANT: {answer}.
```

**Training Strategy.** To jointly train the discriminative loss and generative loss, we calculate the loss as follows. Since the last token of an image is often a padding token, we take all 576 hidden state tokens before the LM head for images and apply average pooling to obtain a single token. For text, we directly take the last toke in the hidden state of MLLM.

During training, We calculate the discriminative loss using the last token from either the end of the text or the image in the MLLM's hidden state. Notably, in an interleaved input sequence with multiple texts or images, we randomly select multiple last tokens from the same sequence to more efficiently utilize the samples.

**Evaluation Data.** For generative tasks, we first evaluate on a wide range of question-answering tasks and some MLLM-oriented comprehension benchmarks, including VQA-v2 [27], GQA [35], VizWiz [28], ScienceQA-IMG [61], TextVQA [76], POPE [51], MME [95], MMBench [58], LLaVA-Bench (In-the-Wild) [56], MM-Vet [99] and. The split of test sets and the evaluation metrics are aligned with those described in VILA[52] and LLaVA [55].

To test the generative ability in interleaving tasks, we use DEMON [47], a comprehensive benchmark that demonstrative instruction following ability, including a wide variety of multi-modal datasets from different fields and scenarios.

For generation tasks, our evaluation encompasses MSCOCO [38] for image-text retrieval, Visual Storytelling (VIST) [32] for interleaved retrieval and Winoground [79] for fine-grained retrieval.

## E.2 Training

We train the parameters for both the LLM and the MLP for embedding the MLLM's hidden state, initializing from VILA [52]. To enhance efficiency for the LLM, we employ LoRA tuning [30] on the $W_q$ and $W_v$ matrices using low-rank adaptation. In our implementation, we set the rank $r = 128$ and $\alpha = 256$. We utilize the AdamW optimizer [60] in conjunction with a cosine learning rate scheduler. The hyperparameters for the AdamW optimizer are configured with a warm-up ratio of 0.03 and a maximum learning rate of $1e - 4$. Training is conducted on 8 x A800 GPUs for approximately 12 hours.

## E.3 Introduction Experiment Details

**(a) WebQA.** The original WebQA contains two types of questions: *"Qcate": "text"* (open-ended questions) and *"Qcate": "YesNo"* (binary judgment questions). For ease of evaluation, we only used the second type. We selected 500 samples of the "YesNo" question type from WebQA [10], each containing one relevant image-text pair and five unrelated image-text pairs. Since the original data provides responses in declarative sentences, we modified the answers of these samples to be either "yes" or "no" by prompting with *"please answer the question in Yes or No."*

We transformed this dataset into a question-answering format. Each question takes the following form (Due to display problems, we have performed line breaks, the same below.):

```
{system-message}. USER: {qustion}\n{image-text pairs}.\n
please answer the question in Yes or No.\n
ASSISTANT: {answer}.
```

Here is a case for one sample in Figure 5:

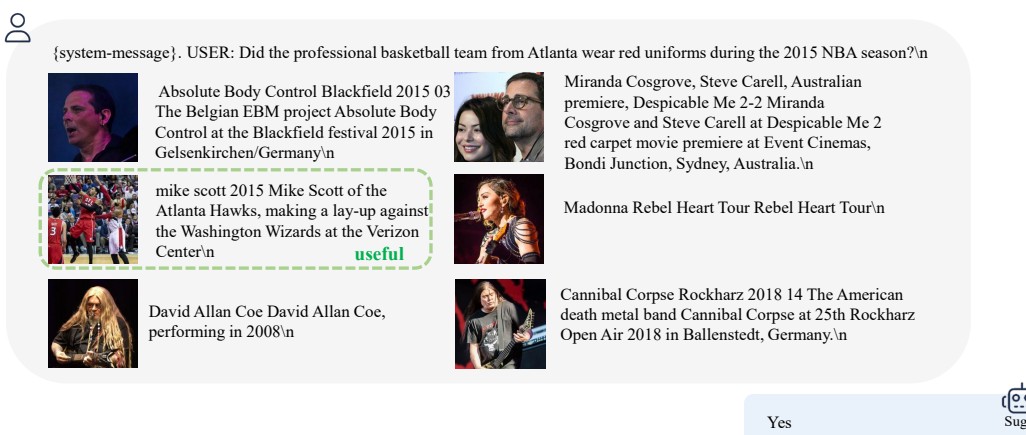

Figure 5: A Case for WebQA. The index for the useful pair is three.

In WebQA, the accuracy roughly forms a "U" shape curve when the relevant image-text pair for a question appears at different positions. While Sugar also shows similar trends, it tends to be more stable overall. Specific numerical results can be found in Table 5.

| Index | 1 | 2 | 3 | 4 | 5 | 6 |
|---|---|---|---|---|---|---|
| VILA | 50.0 | 49.0 | 47.4 | 44.4 | 44.4 | 49.0 |
| **Sugar** | 63.8 | 60.4 | 59.6 | 59.2 | 60.4 | 61.0 |

Table 5: Specific accuracy (%) values displayed on WebQA. The index indicates the position of the useful image-text pair, denoting which position it occupies in the sequence.

**(b) MMVP-VLM Benchmark.** MMVP-VLM [81] contains 30 carefully annotated images in each dimension of capability, with pairs of images being highly similar to each other (as indicated by their high similarity scores in CLIP). To evaluate the discriminative ability of generative models on these finely nuanced images, we transformed this dataset into a question-answering format. Each question takes the following form:

```
{system-message}. USER: First Image:<image>\nSecond Image:<image>\n

which choice meets the first image:\n

A.{data["Statement"]}\nB.{data["Statement2"]}\n.please answer in A or B

ASSISTANT: {answer}.
```

Among them, both Statement 1 and Statement 2, as well as Image 1 and Image 2, are highly similar, with only subtle differences. Furthermore, there is a corresponding relationship between Image 1 and Statement 1, and between Image 2 and Statement 2. We employed a random seed to ensure that the correct answer is equally distributed between option A and option B. The specific values for the experiment are provided in Table 6.

| | Image Size | 🧭 | 🔍 | 🔁 | ↕️ | 📍 | 🎨 | ⚙️ | A | 📷 | Average |
|---|---|---|---|---|---|---|---|---|---|---|---|
| OpenAI ViT-L-14 [73] | $224^2$ | 13.3 | 13.3 | 20.0 | 20.0 | 13.3 | 53.3 | 20.0 | 6.7 | 13.3 | 19.3 |
| OpenAI ViT-L-14 [73] | $336^2$ | 0.0 | 20.0 | 40.0 | 20.0 | 6.7 | 20.0 | 33.3 | 6.7 | 33.3 | 20.0 |
| SigLIP ViT-SO-14 [100] | $224^2$ | 26.7 | 20.0 | 53.3 | 40.0 | 20.0 | **66.7** | 40.0 | 20.0 | **53.3** | 37.8 |
| SigLIP ViT-SO-14 [100] | $384^2$ | 20.0 | 26.7 | 60.0 | 33.3 | 13.3 | **66.7** | 33.3 | 26.7 | **53.3** | 37.0 |
| DFN ViT-H-14 [20] | $224^2$ | 20.0 | 26.7 | 73.3 | 26.7 | 26.7 | **66.7** | 46.7 | 13.3 | **53.3** | 39.3 |
| DFN ViT-H-14 [20] | $378^2$ | 13.3 | 20.0 | 53.3 | 33.3 | 26.7 | **66.7** | 40.0 | 20.0 | 40.0 | 34.8 |
| MetaCLIP ViT-L-14 [88] | $224^2$ | 13.3 | 6.7 | **66.7** | 6.7 | 33.3 | 46.7 | 20.0 | 6.7 | 13.3 | 23.7 |
| MetaCLIP ViT-H-14 [88] | $224^2$ | 6.7 | 13.3 | 60.0 | 13.3 | 6.7 | 53.3 | 26.7 | 13.3 | 33.3 | 25.2 |
| EVA01 ViT-g-14 [78] | $224^2$ | 6.7 | 26.7 | 40.0 | 6.7 | 13.3 | **66.7** | 13.3 | 13.3 | 20.0 | 23.0 |
| EVA02 ViT-bigE-14+ [78] | $224^2$ | 13.3 | 20.0 | **66.7** | 26.7 | 26.7 | **66.7** | 26.7 | 20.0 | 33.3 | 33.3 |
| VILA-7B [52][†] | $336^2$ | 36.7 | 46.7 | 53.3 | 43.3 | 50.0 | 60.0 | 50.0 | 46.7 | 50.0 | 48.5 |
| Sugar[†] | $336^2$ | **56.7** | **50.0** | 63.3 | **50.0** | **60.0** | **66.7** | **56.7** | **63.3** | **53.3** | **57.8** |

Table 6: Performance Comparison of VILA and Various CLIP-Based Models on Different Visual Patterns in MMVP-VLM Benchmark. For most of the visual patterns, all CLIP-based methods show struggle, as evident from the scores. Sugar achieves state-of-the-art performance on the majority of tasks, demonstrating its powerful discriminative ability. We use symbols for visual patterns due to space limit: 🧭: Orientation and Direction, 🔍: Presence of Specific Features, 🔁: State and Condition, ↕️: Quantity and Count, 📍: Positional and Relational Context, 🎨: Color and Appearance, ⚙️: Structural and Physical Characteristics, A: Texts, 📷: Viewpoint and Perspective. [†] indicates that we use question-answering as the test method, instead of dot product.

## E.4 Retrieval-Augmented Generation.

For performing retrieval-augmented generation (RAG), we selected two tasks, namely VizWiz and SQA$^I$, as they provide held-in data not seen during model training. We did not use VQA$^{v2}$, GQA, and VQA$^T$ because their held-in data is a subset of the widely-used LLaVA-1.5 SFT. Benchmarks like POPE, MMB, and others lack held-in data. Therefore, we focused on VizWiz and SQA$^I$ for our experiments. We utilized a mixed set comprising the widely-used LLaVA-1.5 SFT subset and the held-in dataset of the two tasks as the knowledge base and employed different retrieval modules to retrieve relevant knowledge for the MLLM. Similar to common practice, we average the similarity scores for CLIP (We choose CLIP ViT-L/14@336px). For BLIP-2, we compute the similarity using its multimodal token's CLS token. Figure 7 shows some specific retrieval results on test data, demonstrating that Sugar can better integrate information from both images and text, retrieving more similar data as external knowledge.

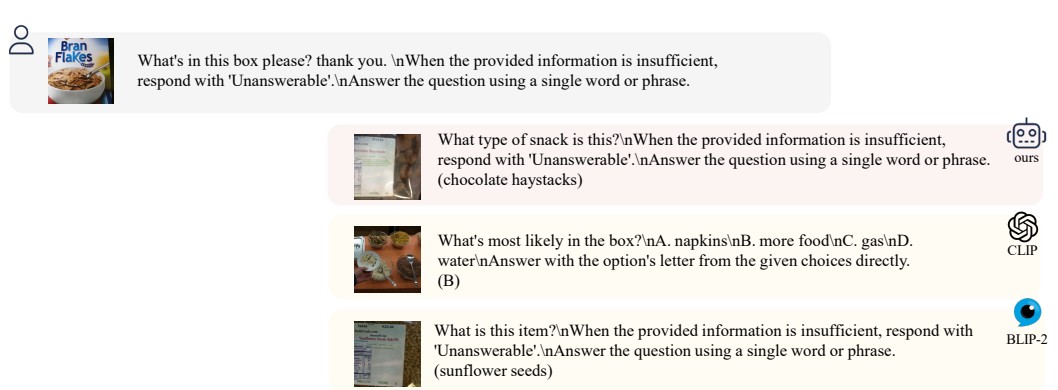

Figure 6: Selected examples from do retrieval-augmented generation.Sugar can retrieve more useful knowledge compared with CLIP and BLIP-2. Inside the parentheses are the answers, note that the When retrieving, we will only retrieve the questions, not the answers, which are shown here for convenience only.

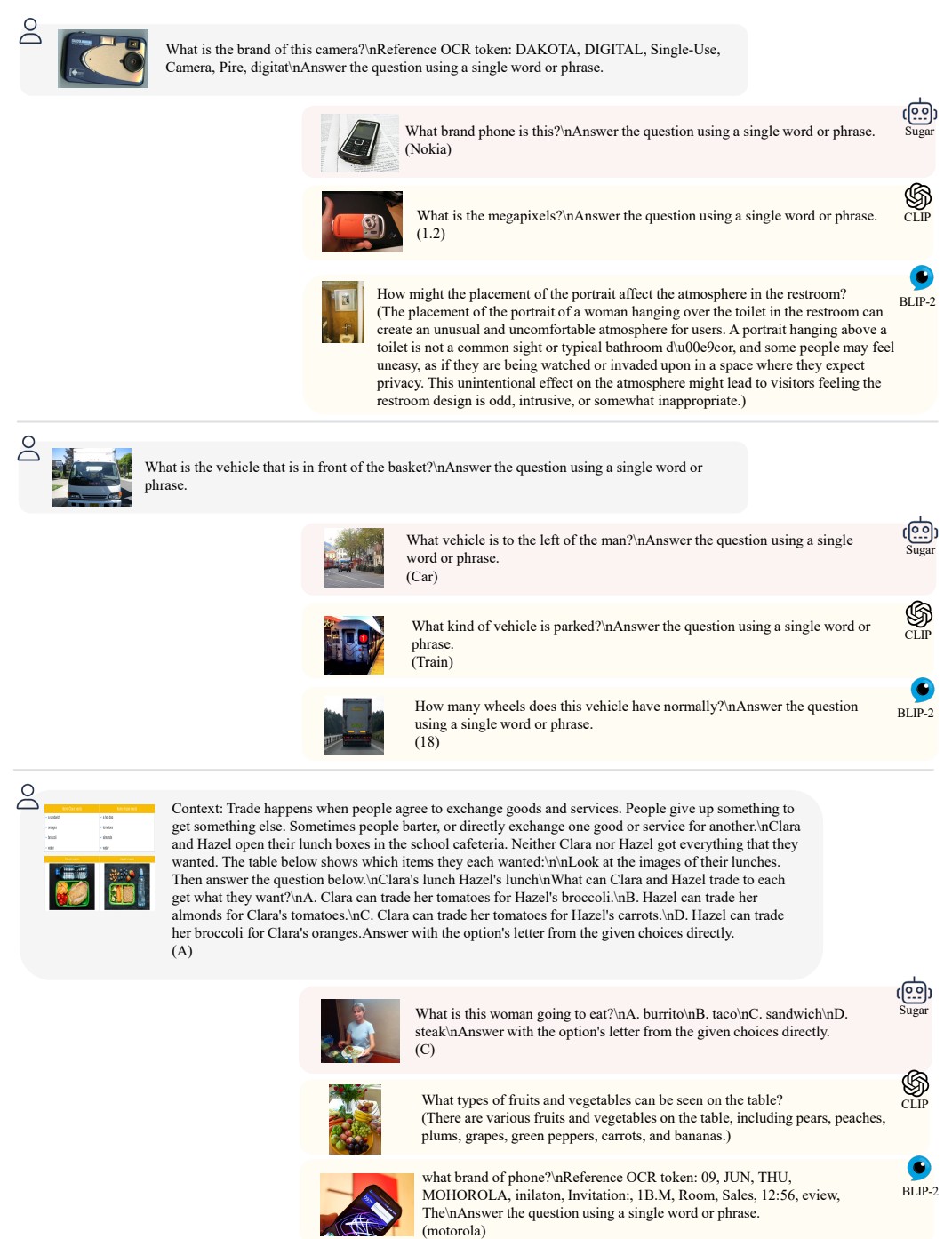

Figure 7: Selected examples from do retrieval-augmented generation (continued for Figure 6).

# F More Results

## F.1 Details of Retrieval

**Image-text Retrieval.** FROMAGe [40] was evaluated on the 5K validation set of MSCOCO 2017. Due to the split method confusion in FROMAGe, we report our image-text retrieval results on MSCOCO val2014's 5K val set following UniIR [84] and the Karpathy split [38]. What's more we then utilize FAISS [37], a powerful library for efficient similarity searches in dense vector spaces, to

index and retrieve candidates. Therefore, the results may exhibit slight differences when compared under identical settings. The results in Table 3(a) are provided for reference only.

**Interleaved Retrieval.** We conduct evaluations across several experimental configurations, following the same setup as FROMAGe [40]. The settings are as follows:

1. Retrieval of the last image given the descriptions of the preceding 5 images. This evaluates models' ability to condition on temporally dependent language.

2. Retrieval of the last image given the descriptions of the preceding 5 images and the 4 preceding images. This assesses models' capability to process interleaved image-and-text context.

**Fine-grained Retrieval.** Winoground [79] is designed to evaluate the ability of vision and language models to perform vision-linguistic compositional reasoning. The task involves matching two images with two captions, where both captions contain an identical set of words/morphemes arranged in different orders. This dataset, meticulously hand-curated by expert annotators, includes a rich set of fine-grained tags to facilitate detailed performance analysis.

## F.2 Quality Results

To analyze Sugar's emergent behaviors and observed weaknesses, we present additional qualitative samples that were not included in the main paper due to space constraints. Please note that for brevity, we have omitted the system prompts and the line breaks after the images for all the quality examples.

We hope these additional results and observations showcase the potential of Sugar in various application areas. In future work, it is important to investigate these emergent behaviors more thoroughly and to understand the underlying mechanisms that enable Sugar to demonstrate such generalization abilities. This will pave the way towards building better MLLMs, including enhancing robustness, reducing biases, and improving the alignment and scope of the learned vision-language representations.

**World Knowledge**: We observe that Sugar can leverage the world knowledge [26] embedded within the LLM to enhance performance on multimodal tasks. For example, as shown in Figure 4, the model understands that during Halloween, people typically dress up in various ways to portray scary, funny, or creative characters, such as ghosts and skeletons.

**Retrieval the Same Sequence at Different Place**: One interesting emergent behavior of Sugar is its ability to retrieve sequences from different positions within the input interleaved sequence, demonstrating flexibility and high sample efficiency, as shown in Figure 4. Unlike CLIP, which requires encoding each sample separately, Sugar can encode sequences of varying lengths for the same multi-modal document in a single forward pass.

What's more, Sugar is capable of both retrieval and generation tasks. Below in Figure 8 are some examples from the VIST dataset.

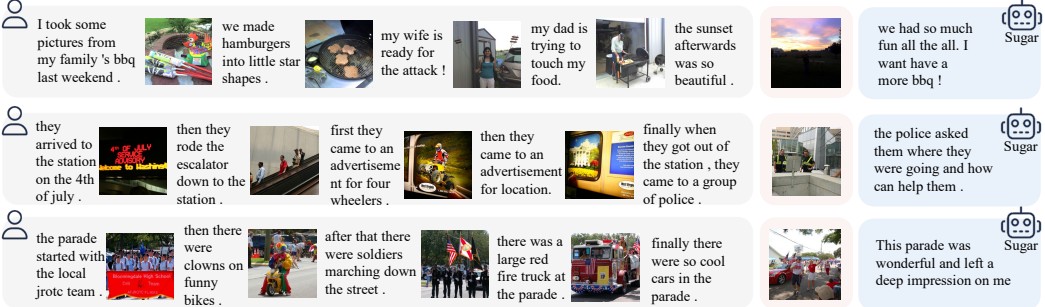

Figure 8: Selected examples for various image-text tasks. The pink background indicates retrieval results, while the blue background indicates generated results.

**Fine-grained Image Discrimination**: As shown in Figure 9, Sugar excels at accurately discerning subtle differences between images and identifying detailed objects and their attributes. VILA, on the other hand, tends to describe the content of the images without pinpointing the precise differences between them. In contrast, Sugar provides more concise and direct answers.

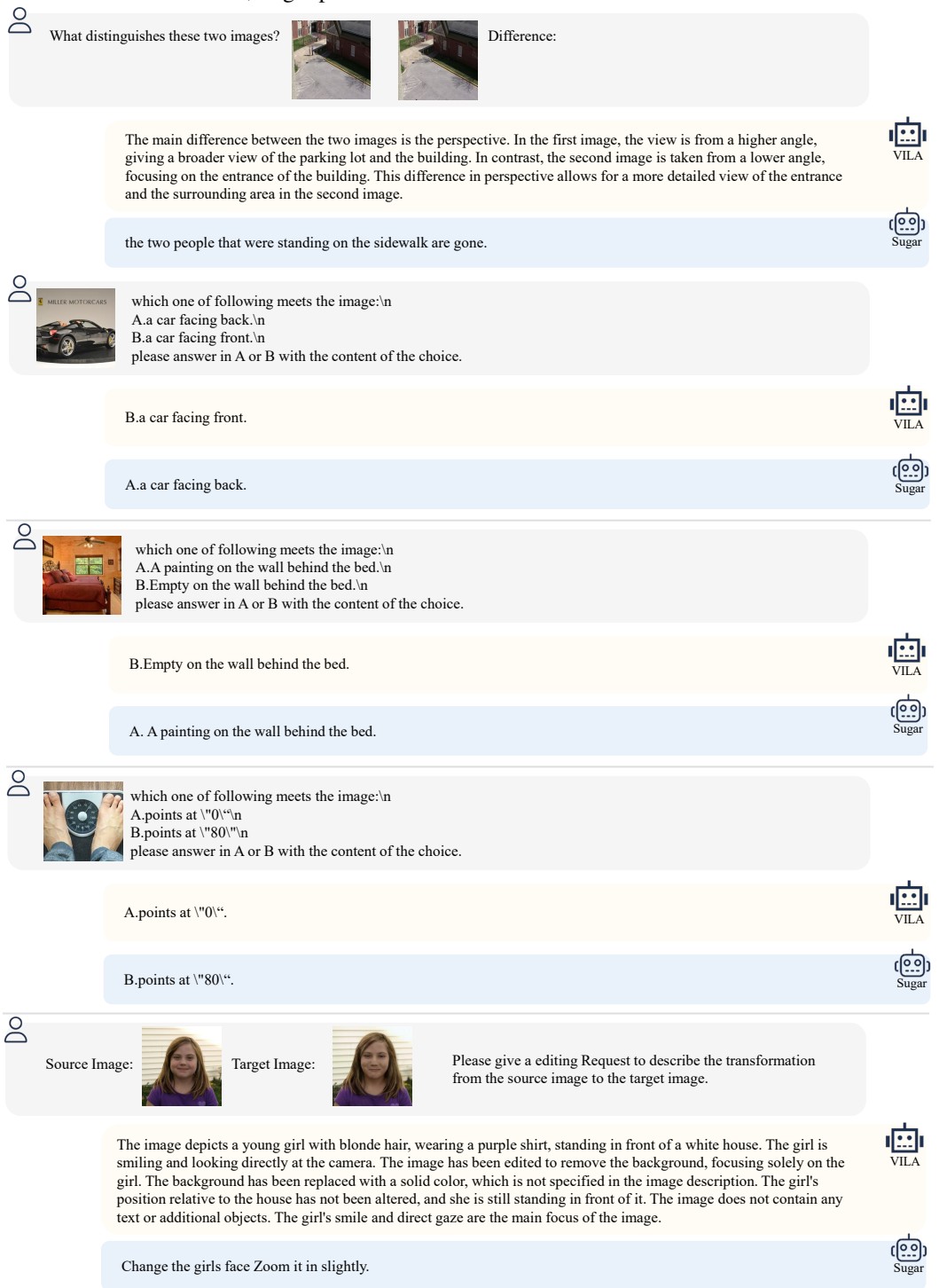

Figure 9: Selected examples. Sugar excels at accurately discerning subtle differences between images and identifying detailed objects and their attributes.

**Style Following**: Sugar exhibits a certain degree of in-context style following capability. As shown in Figure 10, with the aid of external knowledge, Sugar partially adopts the style of retrieved results, resulting in more accurate and detailed answers compared to scenarios without retrieval augmentation.

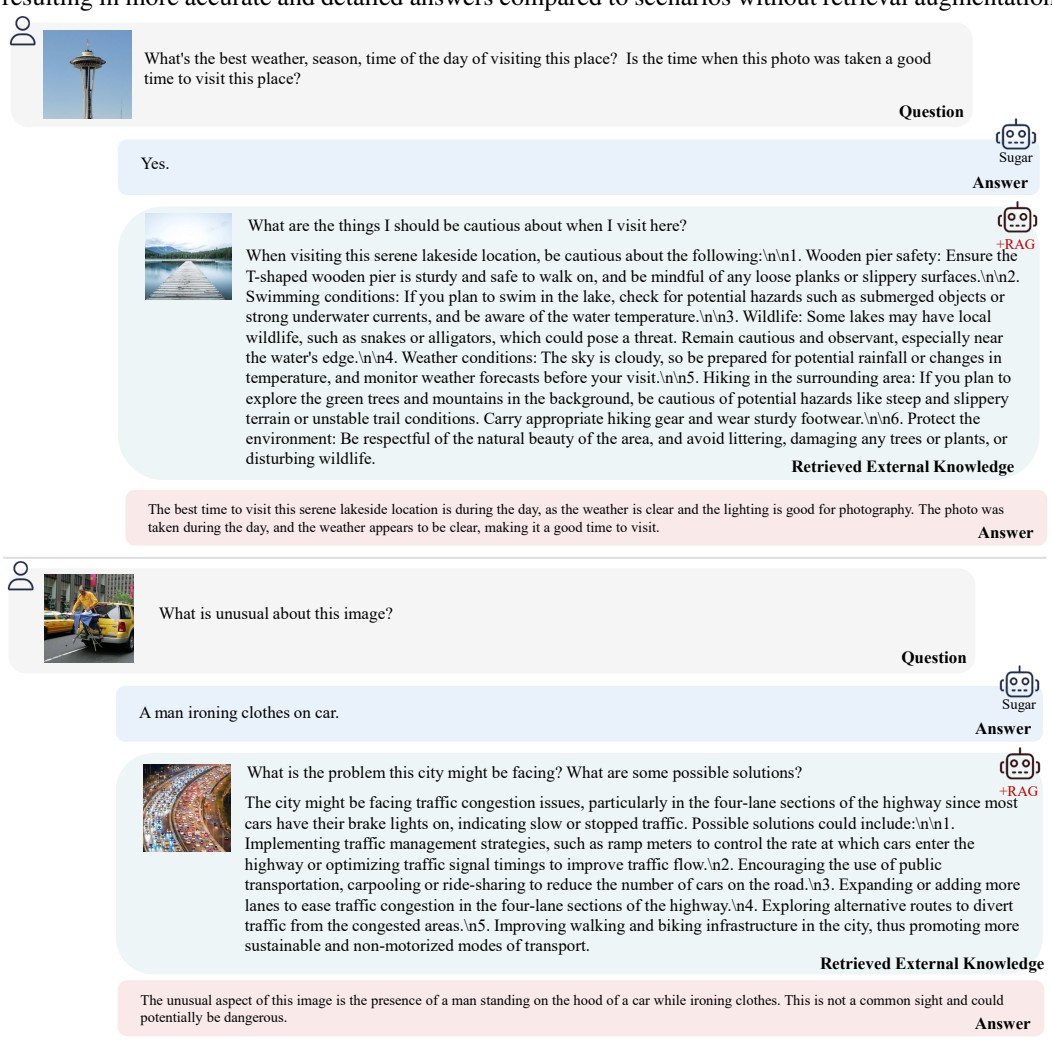

Figure 10: Selected examples from LLaVA-Bench(In-the-wild). Using external knowledge, Sugar partially follows the style of retrieved results, providing more accurate and detailed answers compared to not using retrieval augmentation.

**Interleaved Comprehension**: As demonstrated by results from DEMON [47], Sugar exhibits superior interleaved comprehension capabilities compared to VILA, particularly in tasks requiring fine-grained analysis and an understanding of global context. For instance, in the third example of Figure 11, VILA confuses character names, whereas Sugar maintains narrative coherence while adhering to the style of the preceding text. Similarly, in the third example of Figure 12, VILA provides an irrelevant response, while Sugar delivers a more contextually appropriate answer. Additionally, Figure 13 demonstrates Sugar's ability to effectively capture global information, identifying the relevant images and text within the sequence to provide accurate responses.

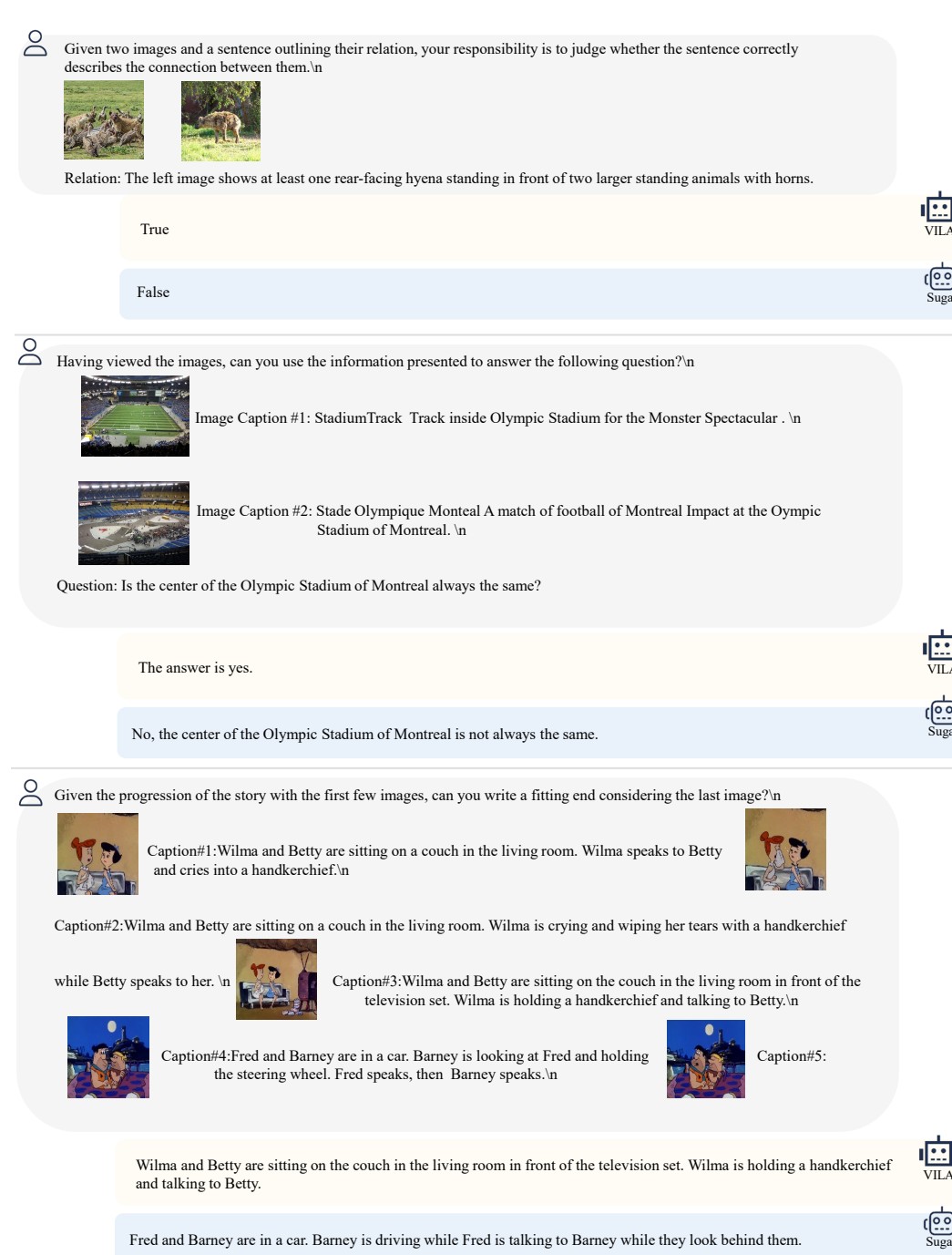

Figure 11: Selected examples for Interleaved Comprehension.

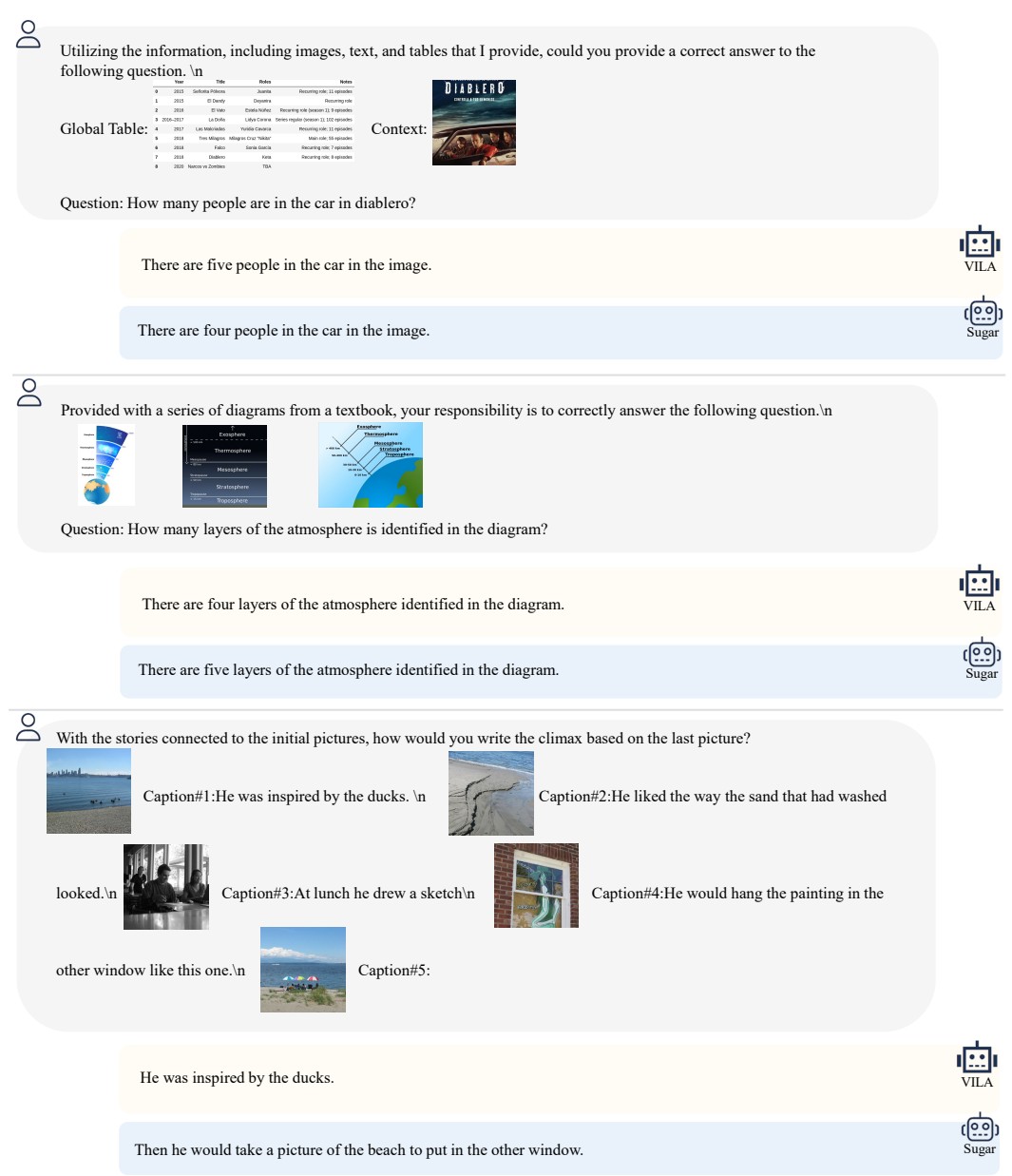

Figure 12: Selected examples for Interleaved Comprehension (continued for Figure 11).

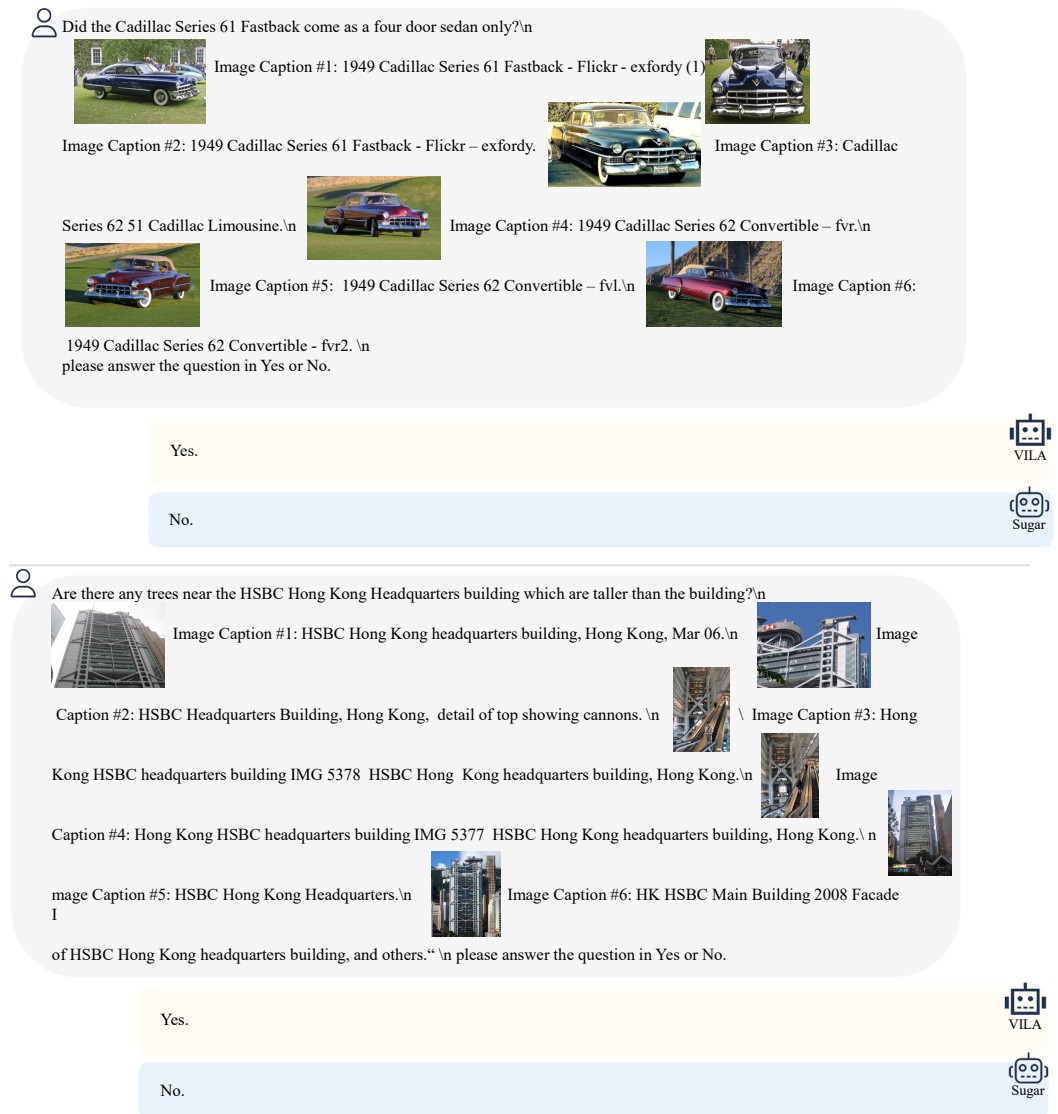

Figure 13: Selected examples for Interleaved Comprehension (continued for Figure 12).

**Sensitivity with Detailed Semantics**: Sugar can address various examples inspired by the Winograd schema [43]. These examples consist of multiple sentences that differ only by a single word, leading to different resolutions of ambiguity. Sugar can accurately match images and text, demonstrating its sensitivity to even minor changes in input prompts. Figure 14 showcases some cases that align with the Winograd schema from Winoground.

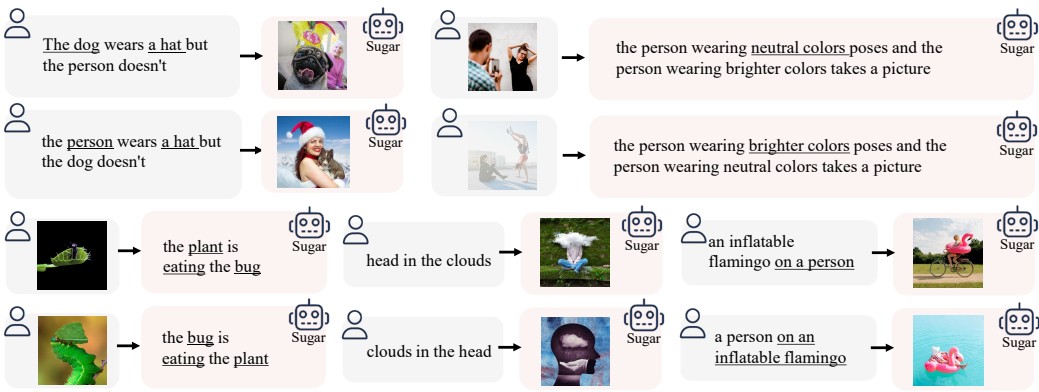

Figure 14: Selected examples from Winoground. Sugar is Sensitivity with Detailed Semantics

# G   Retrieval for Knowledge-based VQA

In this section, we use FVQA [83] and WebQA [10], two knowledge-based VQA datasets, to verify Sugar's effectiveness of combining retrieval and comprehension abilities in a single model, thereby avoiding compatibility issues and suboptimal performance.

Historically, solving FVQA has relied on modeling the knowledge database using a Knowledge Graph [13]. For WebQA, each question is associated with 10-20 knowledge bases, but only one is relevant to the image and caption. FVQA knowledge is textual, whereas WebQA knowledge consists of both text and pictures.

**Implement Details**. In this experiment, we used CLIP ViT-L/14@336px, and both experiments report the ROUGE-L Score.

For FVQA, answers originate from two sources: directly from the image or from the knowledge base. To minimize interference, we only tested questions requiring the knowledge base. We used the following prompt for FVQA: "Please answer the questions based on the pictures. If the reference information is useful, please use it. Otherwise, please ignore the reference information. Reference information: retrieved knowledge <image> question." The *baseline* without retrieval means we did not search for knowledge, but directly input the image and question for the model to answer. *+ CLIP image* means using the image to retrieve knowledge, *+ CLIP text* means using the text to retrieve knowledge, and *+ CLIP average* means using the average annotations of both image and text to retrieve knowledge. For our model, *sugar+rag* indicates the average result obtained using both image and text to retrieve knowledge.

For WebQA, each question has 10-20 negative captions and images. Due to context length limitations in LLaVA and VILA, we could not input all the data, necessitating a retrieval model to extract relevant knowledge. Due to the large dataset size, we randomly selected 1000 samples. For WebQA, *+CLIP image* means providing the positive image and using it to retrieve the most relevant text from the knowledge base, which is then used as input for the model to answer the question. Conversely, *+CLIP text* uses the text to retrieve relevant images. For our model, *sugar+rag* indicates the result obtained using the average similarity score of the aforementioned methods.

|                                | FVQA | WebQA |
|--------------------------------|------|-------|
| LLaVA-1.5-7B                   | 5.9  | /     |
| LLaVA-1.5-7B + CLIP image      | 6.8  | 81.8  |
| LLaVA-1.5-7B + CLIP text       | 7.1  | 79.2  |
| LLaVA-1.5-7B + CLIP (average)  | 7.9  | /     |
| VILA-7B                        | 6.4  | /     |
| VILA-7B + CLIP image           | 9.0  | 80.0  |
| VILA-7B + CLIP text            | 10.2 | 71.2  |
| VILA-7B + CLIP (average)       | 11.0 | /     |
| **Sugar**                      | 6.5  | /     |
| **Sugar + rag**                | **20.7** | **88.7** |

Table 7: Comparison between the independent generator + retriever and Sugar on knowledge-based VQA. '/' indicates not applicable.

**Results**. Based on Table 7, we can observe that while MLLM can answer a small portion of FVQA questions using its internal knowledge, it still requires the support of a retriever for enhanced accuracy. However, the impact of retrieval strategies on the results is inconsistent. For instance, using text retrieval often outperforms image retrieval in FVQA, whereas in WebQA, image retrieval is more effective. Additionally, there are compatibility issues between retrieval strategies and models. For example, in WebQA, VILA is more sensitive to CLIP's retrieval strategy, with fluctuations 3.4 times greater than those of LLaVA-1.5. Our integrated retriever and generator model, however, does not require an additional retriever and avoids the aforementioned optimization and selection issues.

