# OpenReview forum: "Unified Generative and Discriminative Training for Multi-modal Large Language Models"
_NeurIPS.cc/2024/Conference — NeurIPS 2024 poster_

### Official Review · Reviewer_DDMF · 2024-07-10

**Soundness:** 3
**Presentation:** 3
**Contribution:** 3
**Rating:** 6
**Confidence:** 2

**Summary:**

This paper proposes a novel learning paradigm to learn MLLMs based on interleaved image-text corpora.
It introduces a structure-induced training strategy that imposes semantic relationships between input samples and
 the MLLM’s hidden state.
This work apply the dynamic time warping framework to calculate the semantic similarity between different image-text sequences.
Then, a discriminative loss is applied to sequence similarity matrices calculated based on raw inputs and MLLM hidden states.
The framework can also leverage the capabilities of multiple vision and language encoders to more accurately calculate the similarity matrices.
Experiment results show that the new learning paradigm demonstrates good performance on basic multimodal comprehension benchmarks,
complicated multimodal comprehension benchmark DEMON, cross-model information retrieval, and retrieval-augmented generation.

**Strengths:**

1. This paper is well-written and easy to follow.
2. This paper proposes a novel learning paradigm based on interleaved image-text corpora.

**Weaknesses:**

1. This paper did not discussed the impact of including interleaved image-text pairs in MLLM learning. For example, how will it affect the performance on basic visual-language benchmarks (Table 1) and image-text retrieval. Will there be any negative effects?

**Questions:**

1. Can sugar better leverage the multi-modal in-context examples or better understand interleaved image-text content, is there any evaluation for that?
2. What is exactly the amount of interleaved image-text sequences (from MMC4) and image-text pairs (from other datasets) used to train Sugar.
3. What is the context window size of Sugar?

**Limitations:**

The authors have adequately addressed the limitations.

---

> ### Author Rebuttal · Authors · 2024-08-07
>
> We sincerely thank you for your comprehensive comments and constructive advice. We will explain your concern as follows.
>
> > **Q1:** This paper did not discussed the impact of including interleaved image-text pairs in MLLM learning. For example, how will it affect the performance on basic visual-language benchmarks (Table 1) and image-text retrieval. Will there be any negative effects?
>
> **A1:** Thank you for your insightful question. We conducted additional ablation experiments (**Table F of the Rebuttal PDF**) by gradually increasing the MMC4 sampling ratio in basic visual-language benchmarks (Table 1) and image-text retrieval. The results demonstrated the following effects of interleaved image-text pairs on MLLM learning.
>
> **(1.1)** MMC4 does not directly contribute to performance improvement for basic visual-language benchmarks. This is because its long sequence undermines the model's interleaved ability (e.g., VQA dropped from 79.9 to 60.5 when MMC4 radio increased). Additionally, without Sugar, MMC4 is unable to assist in image-text retrieval.
>
> **(1.2)** (a) Delightfully, using a synergistic framework in Sugar can significantly enhance the capability of interleaved text  while maximizing the retention of basic visual-language capabilities as the MMC4 data increases. Moreover, it will facilitate certain tasks that require distinguishing between multiple images or texts, such as POPE (85.5 to 86.4）and multi-image VQA(from 38.4 to 39.5). (b) Only Sugar can simultaneously achieve image-text retrieval. With the increase in MMC4 data, it can significantly improve its performance in complex retrieval tasks (increasing from 14.4 to 23.2) while maintaining competitive performance in general retrieval tasks.
>
> *Please refer to Table F of the Rebuttal PDF for the Results. The table below is the same as in the PDF*
>
> > **Q2:** Can Sugar better leverage the multi-modal in-context examples or better understand interleaved image-text content, is there any evaluation for that?
>
> **A2:** Thank you for your constructive suggestions. Following your recommendations, we have conducted additional experiments on 5 benchmarks across 3 task types to further evaluate the effectiveness of our method in **Table E**.
>
> To effectively answer **multi-image QA**, the model must clearly distinguish between different images and understand their common global information. In **in-context interaction**, the model must fully comprehend the preceding image-text dialogue to provide reasonable responses. For **visual prompt comprehension**, the model needs to meticulously identify visual cues in the images to answer questions. Our model achieved the best results across all three task types, especially in VIST-SIS, which involves 5 rounds of interleaved image-text dialogue.
>
> *Table E Comparison with baseline in 5 challenging tasks*
>
> |              | **Multi-image VQA** |          | **In-context Interaction** |          | **Visual Prompt** |
> | ------------ | ------------------- | -------- | -------------------------- | -------- | ----------------- |
> |              | SEED                | Mantis   | Visual Dialog              | VIST-SIS | BLINK             |
> | LLaVA-1.5-7B | 58.6                | 31.3     | 53.7                       | 14.1     | 37.1              |
> | VILA-7B      | 61.1                | 38.4     | 58.5                       | 17.4     | 39.2              |
> | Sugar        | **63.6**            | **41.0** | **62.5**                   | **28.2** | **42.2**          |
>
> We also present the results in Table E of the Rebuttal PDF.
>
> > **Q3:** What is exactly the amount of interleaved image-text sequences (from MMC4) and image-text pairs (from other datasets) used to train Sugar.
>
> **A3:**  Thank you for your question. We apologize if our manuscript gave the impression that the training recipe for Sugar was unclear. The specific data usage and comparisons are detailed in the **Table A**. We did **not** use any additional data beyond what baseline used.
>
> We utilized a **limited dataset** in comparison to VILA and LLaVA-1.5. Nonetheless, we achieved competitive outcomes and acquired additional retrieval and retrieval-augmented capabilities. By employing an equivalent data volume and integrating the 300K SFT data used by VILA， we observed improvements in 12 multi-modal comprehension capabilities (**Table B,C**), showcasing the model's scalability.
>
> > **Q4:** What is the context window size of Sugar?
>
> **A4:**  Thank you for your insightful query! Due to the context length limitation of Vicuna-7B, which is **4096** tokens. So, Sugar cannot handle very long multimedia documents.  However, we believe that the mutual enhancement of generative and discriminative capabilities demonstrated by Sugar, as well as the exploration of integrating retrieval and generation within a single model, are beneficial. Your question is valuable and will guide our future improvements and research directions.
>
> Thank you once again for your recognition and constructive suggestions, which have been instrumental in enhancing the quality of our research!

---

> ### Author Response · Authors · 2024-08-07
> **Table B**
>
> Table B: Comparison with baseline on 11 visual-language benchmark with equitable data volume.
>
> |                     | VQA       | GQA        | VizWiz   | SQAI     | VQAT     | POPE     | MMEP       | MMEC      | MMB      | LLaVAWd  | MM-Vet   |
> | ------------------- | --------- | ---------- | -------- | -------- | -------- | -------- | ---------- | --------- | -------- | -------- | -------- |
> | LLaVA-1.5-7B        | 78.5     | 62.0      | 50.0     | 66.8     | 58.2     | 85.9     | 1510.7     | –         | 64.3     | 49.0     | 30.5     |
> | VILA-7B             | 79.9    | 62.3      | 57.8     | 68.2     | 64.4     | 85.5     | 1533.0     | 296.1     | 68.9     | 70.0     | 34.9     |
> | Sugar               | 76.0    | 58.7     | 60.4     | 69.4     | 57.5     | 86.6     | 1550.8     | 300.0     | 64.9     | 75.6     | 31.3     |
> | with equitable data volume | **80.2** | **63.1** | **61.0** | **72.1** | **65.1** | **87.2** | **1550.0** | **309.0** | **69.3** | **76.4** | **36.8** |
>
> *We also present the results in Table B of the Rebuttal PDF.*

---

> ### Author Response · Authors · 2024-08-07
> **Table E**
>
> Table E: Comparison with baseline in 5 challenging tasks.
> |                     | **Multi-image VQA** |          | **In-context Interaction** |          | **Visual Prompt** |
> | ------------------- | ------------------- | -------- | -------------------------- | -------- | ------------------ |
> |                     | SEED                | Mantis   | Visual Dialog              | VIST-SIS | BLINK              |
> | LLaVA-1.5-7B        | 58.6                | 31.3     | 53.7                       | 14.1     | 37.1               |
> | VILA-7B             | 61.1                | 38.4     | 58.5                     | 17.4     | 39.2               |
> | Sugar | **63.6**            | **41.0** | **62.5**                 | **28.2** | **42.2**           |
>
> *We also present the results in Table E of the Rebuttal PDF.*

---

> ### Author Response · Authors · 2024-08-07
> **Table F**
>
> Table F: Ablation study on the effects of MMC4. We evaluated basic visual-language benchmarks, including VQA and GQA. For image-text retrieval, we reported MSCOCO R@5 for both text-to-image(t2i) and image-to-text (i2t). Additionally, we included Mantis for multi-image VQA and VIST R@5 for complex retrieval tasks.
> |                             |                   | basic visual-language benchmarks |      |        |      |      |      |      | image-text retrieval |                | multi-image VQA | complex retrieval |
> | --------------------------- | ----------------- | -------------------------------- | ---- | ------ | ---- | ---- | ---- | ---- | -------------------- | -------------- | --------------- | ----------------- |
> |                             | MMC4 sample ratio | VQA                              | GQA  | VizWiz | SQAI | VQAT | POPE | MMB  | t2i    | i2t | Mantis          | VIST      |
> | VILA-7B                     |                   | 79.9                             | 62.3 | 57.8   | 68.2 | 64.4 | 85.5 | 68.9 | /                    | /              | 38.4            | /                 |
> | + directly fine-tune        | 25%               | 72.3                             | 59.9 | 53.5   | 64.1 | 58.7 | 84.8 | 63.4 | /                    | /              | 38.5            | /                 |
> |                             | 50%               | 67.8                             | 54.7 | 49.1   | 58.9 | 51.8 | 82.8 | 59.7 | /                    | /              | 38.7            | /                 |
> |                             | 75%               | 60.5                             | 49.1 | 39.3   | 43.4 | 43.2 | 81.7 | 53.2 | /                    | /              | 38.2            | /                 |
> | + fine-tune with our method | 25%               | 78.7                             | 60.2 | 60.6   | 70.1 | 62.1 | 85.9 | 66.1 | 40.3                 | 36.2           | 38.7            | 14.4              |
> |                             | 50%               | 75.9                             | 57.9 | 59.2   | 65.9 | 57.0 | 86.4 | 62.3 | 40.7                 | 36.3           | 39.0            | 19.1              |
> |                             | 75%               | 73.7                             | 55.1 | 57.1   | 62.3 | 51.7 | 85.7 | 59.7 | 40.2                 | 36.0           | 39.5            | 23.2              |
>
> *We also present the results in Table F of the Rebuttal PDF.*

---

> ### Author Response · Authors · 2024-08-11
> **Looking Forward to Your Reply**
>
> Dear Reviewer DDMF,
>
> Thank you for the time and effort you have dedicated to reviewing our submission. We hope we have addressed the concerns raised in your initial reviews and eagerly await your thoughts and further guidance to refine our work. As the author-reviewer discussion period for NeurIPS 2024 will be over soon, please let us know if you require any additional information or clarification from our end. We are open to engage in further discussions to enhance our submission. Thank you!

---

> > ### Comment · Reviewer_DDMF · 2024-08-13
> >
> > My concerns are well resolved in the rebuttal, and I would like to raise my rating to 6.

---

> > > ### Author Response · Authors · 2024-08-13
> > >
> > > Thank you for your support of our work. Your valuable feedback has made our work better!

---

### Official Review · Reviewer_Lcdg · 2024-07-12

**Soundness:** 3
**Presentation:** 3
**Contribution:** 3
**Rating:** 6
**Confidence:** 4

**Summary:**

The paper addresses the limitations of Vision-Language Models (VLMs) by proposing a unified approach that combines generative and discriminative training paradigms. This new method leverages interleaved image-text sequences and introduces a structure-induced training strategy. It aims to enhance the MLLM's ability to capture global semantics and fine-grained details, effectively balancing generative and discriminative tasks. The approach uses dynamic sequence alignment within the Dynamic Time Warping framework and integrates a novel kernel for fine-grained semantic differentiation. Extensive experiments demonstrate that this method achieves state-of-the-art results in various generative and discriminative tasks.

**Strengths:**

- The paper introduces a novel method that successfully integrates generative and discriminative training paradigms, addressing the weaknesses inherent in each when used independently.
- The authors clearly articulate the challenges faced by existing VLMs and provide a well-defined solution.

**Weaknesses:**

- While the paper shows impressive results, there is limited discussion on the potential limitations and areas where the model might underperform.
- The paper primarily focuses on specific benchmarks. It would be beneficial to discuss how well the proposed method generalizes to other types of vision-language tasks not covered in the experiments.

**Questions:**

- Can you provide more detailed ablation studies to understand the contribution of each component of the proposed method, such as the dynamic sequence alignment and the GAK?

**Limitations:**

- Considering the connection with real-world scenarios is necessary. The authors need to discuss: what requirements might the tasks introduced in the paper have in real-world scenarios?

---

> ### Author Rebuttal · Authors · 2024-08-07
>
> We sincerely thank you for the valuable comments and we will explain your concern as follows.
>
> > **Q1:** While the paper shows impressive results, there is limited discussion on the potential limitations and areas where the model might underperform.
>
> **A1**: Thank you very much for your valuable question! Our potential limitations are as follows:
>
> **(1.1)** Firstly, the context length limitation of Vicuna, capped at 4096 tokens, poses a significant challenge. This restricts Sugar's ability to handle very long multimedia documents.
>
> **(1.2)** Moreover, real-world applications often present varied and dynamic data, which might not align perfectly with the model's training data. This discrepancy could lead to suboptimal performance in diverse and unforeseen contexts.
>
> **(1.3)** The limitation of our model lies in its reliance on the existing Vicuna and CLIP encoder,  which is suboptimal and somewhat limits the comprehension ability *[Karamcheti et al., 2024]*. For example, there are some hallucinations.
>
> *[Karamcheti et al., 2024]* Prismatic VLMS: Investigating the Design Space of Visually-Conditioned Language Models. ICML, 2024.
>
> > **Q2:** The paper primarily focuses on specific benchmarks. It would be beneficial to discuss how well the proposed method generalizes to other types of vision-language tasks not covered in the experiments.
>
> **A2:** Thank you for your suggestion. In the original manuscript, we primarily used the single-image QA benchmarks from LLaVA-1.5 to evaluate our model's multi-modal comprehension capabilities. To provide a more comprehensive evaluation of our model, we have supplemented our method with 5 benchmarks in **Table E of the Rebuttal PDF**.
>
> To answer **multi-image QA**, the model must clearly distinguish between different images and understand their common global information. In **in-context interaction**, the model must fully comprehend the preceding image-text dialogue to provide reasonable responses (Visual Dialog and VIST-SIS, involve 10 and 5 rounds of dialogue respectively). For **visual prompt comprehension**, the model must meticulously identify visual cues in the images to answer questions. Our model achieved the best results across all three task types.
>
> > **Q3:** Can you provide more detailed ablation studies to understand the contribution of each component of the proposed method, such as the dynamic sequence alignment and the GAK?
>
> **A3:** To test the effectiveness of our method, we introduced two variants of GAK: GMin and GMax, and examined the impact of the hyperparameter $\delta$. To avoid interference from additional data, we explored the impact of different parameters without using any extra task-specific data. We tested two variants of the kernel function:
>
> GMin kernel: $
> K_{GMin}(\mathbf{x}, \mathbf{y}) =\min_{\pi \in \mathcal{A}(\mathbf{x}, \mathbf{y})} \prod_{i=1}^{|\pi|} e^{-\phi_\sigma}  \in (0, 1]
> $,
>
> GMax kernel$:
> K_{GMax}(\mathbf{x}, \mathbf{y}) =\max_{\pi \in \mathcal{A}(\mathbf{x}, \mathbf{y})} \prod_{i=1}^{|\pi|} e^{-\phi_\sigma}  \in (0, 1]
> $.
>
> Based on the experimental results in **Table G**, we found that:
>
> 1. GAK is relatively stable and not very sensitive to the hyperparameter δ.
> 2. GAK generally performs better compared to GMin and GMax.
> 3. Tasks requiring global information, such as VIST and KGQA, and those needing detailed semantics, like Winground, are the most sensitive to changes in the method.
>
> > **Q4:** Considering the connection with real-world scenarios is necessary. The authors need to discuss: what requirements might the tasks introduced in the paper have in real-world scenarios?
>
> **A4:**  Thank you for your insightful question. Below, we outline two valuable real-world applications:
>
> **(4.1) Complex information retrieval.** Real-world information retrieval requires models to search through **interleaved multi-modal sequences** , as articles, websites, and books are often composed of extensive interwoven image and text sequences. Current retrievers like CLIP and Sentence BERT are designed to handle single images or texts and struggle with complex interleaved retrieval tasks. In Section 4.4, we tested complex retrieval scenarios and achieved promising results. Adapting to more complex scenarios is an important direction for future research.
>
> **(4.2) Complex comprehension tasks**, such as the 5 new tasks mentioned in Q2, which respectively require capturing global semantics and detailed semantic differentiation.
>
> **(4.3) Knowledge retrieval for answering questions.** In reality, many questions require external knowledge for accurate answers. For tasks such as FVQA and WebQA, traditional methods first require the retriever to find relevant knowledge in a knowledge base, and then the generator to formulate the answer. The final performance, therefore, depends on both the retriever and generator models, highlighting **compatibility issues** and **sub-optimal issues** between them. Based on the **Table D**, we can observe the following:
>
> **(4.3.1)** MLLM can answer a small portion of FVQA questions using internal knowledge, but it still requires some knowledge from a retriever for accurate responses.
>
> **(4.3.2)** The impact of retrieval strategies on the results is inconsistent. For instance, text-based retrieval strategies often outperform image-based ones in FVQA, whereas in WebQA, image-based retrieval is more effective.
>
> **(4.3.3)** There are also compatibility issues between retriever and models. For example, in WebQA, VILA is more sensitive to retrieval strategy, exhibiting fluctuations 3.4 times greater than those of LLaVA-1.5.
>
> **(4.3.4)** Our integrated retriever and generator model does not require additional retrievers, thereby avoiding the optimization and selection issues mentioned above.
>
> We hope this clarifies your concerns. We are committed to thoroughly incorporating your suggestions in the next version of the paper. Thank you once again for your excellent feedback.

---

> ### Author Response · Authors · 2024-08-07
> **Table B**
>
> Table B: Comparison with baseline on 11 visual-language benchmark with equitable data volume.
>
> |                     | VQA       | GQA        | VizWiz   | SQAI     | VQAT     | POPE     | MMEP       | MMEC      | MMB      | LLaVAWd  | MM-Vet   |
> | ------------------- | --------- | ---------- | -------- | -------- | -------- | -------- | ---------- | --------- | -------- | -------- | -------- |
> | LLaVA-1.5-7B        | 78.5     | 62.0      | 50.0     | 66.8     | 58.2     | 85.9     | 1510.7     | –         | 64.3     | 49.0     | 30.5     |
> | VILA-7B             | 79.9    | 62.3      | 57.8     | 68.2     | 64.4     | 85.5     | 1533.0     | 296.1     | 68.9     | 70.0     | 34.9     |
> | Sugar               | 76.0    | 58.7     | 60.4     | 69.4     | 57.5     | 86.6     | 1550.8     | 300.0     | 64.9     | 75.6     | 31.3     |
> | with equitable data volume | **80.2** | **63.1** | **61.0** | **72.1** | **65.1** | **87.2** | **1550.0** | **309.0** | **69.3** | **76.4** | **36.8** |
>
> *We also present the results in Table B of the Rebuttal PDF.*

---

> ### Author Response · Authors · 2024-08-07
> **Table C**
>
> Table C: DEMON benchmark with equitable data volume.
> |                            | MMD      | VST      | VRI      | MMC      | KGQA     | TRQA     | MMR      |
> | -------------------------- | -------- | -------- | -------- | -------- | -------- | -------- | -------- |
> | LLaVA-1.5-7B               | 37.5     | 25.2     | 25.9     | 22.2     | 48.6     | 44.9     | 50.3     |
> | VILA-7B                    | 47.8     | 25.8     | 13.2     | 17.2     | 60.1     | 42.1     | 50.5     |
> | Sugar                      | 51.8     | 34.3     | 32.3     | 16.8     | 64.4     | 65.9     | 51.7     |
> | with equitable data volume | **54.1** | **34.7** | **34.2** | **23.1** | **65.0** | **68.1** | **53.2** |
>
> *We also present the results in Table C of the Rebuttal PDF.*

---

> ### Author Response · Authors · 2024-08-07
> **Table D**
>
> Table D: Knowledge-based VQA.
> |                               | FVQA     | WebQA    |
> | ----------------------------- | -------- | -------- |
> | LLaVA-1.5-7B                  | 5.9      | /        |
> | LLaVA-1.5-7B + CLIP image     | 6.8      | 81.8     |
> | LLaVA-1.5-7B + CLIP text      | 7.1      | 79.2     |
> | LLaVA-1.5-7B + CLIP (average) | 7.9      | /        |
> | VILA-7B                       | 6.4      | /        |
> | VILA-7B + CLIP image          | 9.0      | 80.0     |
> | VILA-7B + CLIP text           | 10.2     | 71.2     |
> | VILA-7B + CLIP (average)      | 11.0     | /        |
> | **Sugar**                     | **6.5**  | /        |
> | **Sugar + rag**               | **20.7** | **88.7** |
>
> *We also present the results in Table D of the Rebuttal PDF.*

---

> ### Author Response · Authors · 2024-08-07
> **Table E**
>
> Table E: Comparison with baseline in 5 challenging tasks.
> |                     | **Multi-image VQA** |          | **In-context Interaction** |          | **Visual Prompt** |
> | ------------------- | ------------------- | -------- | -------------------------- | -------- | ------------------ |
> |                     | SEED                | Mantis   | Visual Dialog              | VIST-SIS | BLINK              |
> | LLaVA-1.5-7B        | 58.6                | 31.3     | 53.7                       | 14.1     | 37.1               |
> | VILA-7B             | 61.1                | 38.4     | 58.5                     | 17.4     | 39.2               |
> | Sugar | **63.6**            | **41.0** | **62.5**                 | **28.2** | **42.2**           |
>
> *We also present the results in Table E of the Rebuttal PDF.*

---

> ### Author Response · Authors · 2024-08-13
> **Awaiting Your Feedback**
>
> Dear Reviewer Lcdg,
>
> Thank you once again for reviewing our submission. We are deeply grateful for your recognition of the value of our work, particularly our proposal of the unified approach that combines generative and discriminative training paradigms to address the challenges of global semantic understanding and detailed semantic capture in existing VLMs. In response to your thoughtful suggestions regarding performance across various vision-language task types and real-world applications, we have conducted additional experiments on 5 more challenging benchmarks, as well as a knowledge-based VQA task of practical significance. These efforts further demonstrate the effectiveness and practical value of our method.
>
> Over the past few days, we have carefully addressed the concerns of other reviewers. They have kindly raised their scores and agreed that our approach not only addresses the limitations of VLMs and enhances comprehension capabilities, but also offers emergent insights into complex retrieval tasks and the unified nature of generative and retrieval abilities, which could inspire future work.
>
> As the author-reviewer discussion period for NeurIPS 2024 is drawing to a close, please feel free to let us know if you have any remaining concerns or questions. We greatly appreciate your feedback and look forward to any further insights you may have.

---

### Official Review · Reviewer_4VP3 · 2024-07-13

**Soundness:** 2
**Presentation:** 2
**Contribution:** 2
**Rating:** 5
**Confidence:** 4

**Summary:**

This paper proposed a method for unifying generative training and discriminative training of multi-modal LLMs. Generative training mainly uses auto-regressive formulation while discriminative training mainly performs contrastive representation matching. The goal of this paper is trying to use discriminative training to improve multi-modal LLMs.

The paper unifies generative training and discriminative training by introducing a Dynamic Sequence Alignment module which aligns similar text and image data on the hidden states of a multi-modal LLM. In addition, Detailed Semantics Modeling is proposed to effectively distinguish detailed semantics.

The paper conducts evaluation on a wide range of benchmarks.

**Strengths:**

The motivation is clear and the paper is easy to follow.

The concept of unifying generative training and discriminative training is interesting.

**Weaknesses:**

It's unclear what is dynamic time warping framework.

The performance improvement of the proposed method sugar is not significant. Compared with some baselines, such as VILA and LLaVA-1.5, Sugar performs worse than them on many tasks, as shown in Table 1. This raises concerns about the effectiveness of the proposed method.

This could be meaningless to align a visual token and a text token in an MLLM model since the LLM is trained with next-token prediction instead of contrastive learning like CLIP. The current token is conditioned on previous tokens. I can't think of a reasonable explanation for this mechanism. It **could be** meaningful to align visual tokens. In addition, the experiment results also suggest that this method is not effective as expected.

What is the evaluation protocol?  Does Sugar train on each benchmark first then evaluate or directly zero-shot evaluation? In the former case, will Sugar lose generative ability after training with discriminative task data?

**Questions:**

What is the training recipe of the proposed method?

**Limitations:**

-

---

> ### Author Rebuttal · Authors · 2024-08-07
>
> We sincerely appreciate your constructive and insightful comments. We will explain your concerns point by point.
>
> First, we must clarify that we only used **a very limited amount of data** (see **Table A of the Rebuttal PDF**). Despite this, we have effectively integrated the generative and discriminative paradigms, alleviating the challenges remain in generative paradigms and collaterally enabling our model to tackle many other tasks. Such as complex retrieval tasks (Section 4.4), multi-modal retrieval-augmented generation (Section 4.5), and knowledge-based VQA in one model.
>
> > **Q1**: It's unclear what is dynamic time warping framework.
>
> **A1:**  Thank you for your question!  We apologize for not clearly explaining our method and the dynamic time warping framework. Below, we will elaborate on DTW and our contributions.
>
> **(1.1)** The Dynamic Time Warping (DTW) framework is a dynamic programming-based method for calculating the similarity of unequal length sequences, commonly used in text information retrieval *[Müller M., 2007]* *[Cuturi M, Blondel M., 2017]* and sequence matching problems *[Cao et al., 2020]*.
>
> **(1.2)** To enable the current separated generative and discriminative paradigms to achieve synergistic gains, we primarily face two challenges: **(a)** comprehensively capturing the global semantics for interleaved sequences of unequal length; **(b)** keenly differentiating the detailed semantics.
>
> **(1.3)** We explicitly designed two corresponding modules in Sugar to tackle these challenges: **(a)** formulate the relationship between interleaved sequences as a dynamic sequence alignment problem within the DTW framework;  **(b)** integrate a novel kernel into the framework to leverage the strengths of various discriminative pre-trained models. The ablation experiments validate the effectiveness of our approach (Section 4.5 and Table G).
>
> *[Müller, 2007]* Dynamic Time Warping. Information Retrieval for Music and Motion.
>
> *[Cuturi & Blondel, 2017]* Soft-DTW: A Differentiable Loss Function for Time-Series. ICML 2017.
>
> *[Cao et al., 2020]* Few-Shot Video Classification via Temporal Alignment. CVPR 2020.
>
> > **Q2**: The performance improvement of the proposed method Sugar is not significant. Compared with some baselines, such as VILA and LLaVA-1.5, Sugar performs worse than them on many tasks, as shown in Table 1. This raises concerns about the effectiveness of the proposed method.
>
> **A2:** Thank you for raising an important concern. We will address your question from the following three aspects:
>
> **(2.1)** We used a very **limited amount of data** compared to VILA. Despite this, we achieved competitive results and gained additional retrieval and retrieval-augmented capabilities. By using an equitable data volume (incorporating 300K SFT data used by VILA), we observed enhancements in 11 comprehension benchmarks of Table 1 (**Table B,C**), demonstrating our model's scalability.
>
> **(2.2)** In the original manuscript Table 1, we primarily used the single-image QA benchmarks. To provide a more comprehensive assessment of our model, we have supplemented our method with 5 new, **more challenging** benchmarks, showcasing our superiority (see **Table E**). To answer **multi-image QA**, the model must clearly distinguish between different images and understand their shared global information. For **in-context interaction**, it must fully comprehend the preceding multiple image-text dialogue to provide reasonable responses. For **visual prompt comprehension**, the model needs to accurately identify visual cues in the images to answer questions. Our model achieved the best results across all three task types.
>
> > **Q3**: This could be meaningless to align a visual token and a text token in an MLLM model since the LLM is trained with next-token prediction instead ... (omitted)
>
> **A3:**  We apologize if our manuscript gave the impression that the method of Sugar was unclear.
>
> **(3.1)**  Simply using contrastive learning methods like CLIP can only calculate the similarity between single images and texts. However, we **consider the interleaved image-text sequence as the general format** of input samples. This means that the number of tokens in different sequences is of unequal length. If we simply use contrastive learning like CLIP, the similarity of interleaved sequences would lose global information.
>
> **(3.2)**  Hence, we select the final token of a sequence to represent this input interleaved sequence. As you mentioned, *'the current token is conditioned on previous tokens*' , the final token can integrate global information from the preceding content. In this way, we can modulate the hidden states of the MLLM by leveraging the semantic relationships between interleaved input sequences, thereby ensuring consistency between hidden states and encouraging the MLLM to capture the global semantics. Additionally, we introduced a novel kernel to provide more detailed signals, enhancing the ability to differentiate fine-grained semantics.
>
> **(3.3)** The effectiveness of our method is demonstrated through experiments on 11 comprehension benchmarks and others VILA can't do (**Table B,C,D,E**).
>
> > **Q4:** What is the evaluation protocol? Does Sugar train on each benchmark first then evaluate or directly zero-shot evaluation? In the former case, will Sugar lose generative ability after training with discriminative task data?
>
> **A4:** All the experiments in our paper are **zero-shot**. We have merely introduced a structure-induced training strategy within the generative paradigm and subsequently conducted zero-shot evaluations for each task.
>
> > **Q5:** What is the training recipe of the proposed method?
>
> **A5:**  We apologize if our manuscript suggested that Sugar's training method was unclear. The specific data we used is detailed in **Table A of the Rebuttal PDF**.
>
> We hope the above discussion addresses your concerns. The discussion remains open, and we always welcome your review.

---

> ### Author Response · Authors · 2024-08-07
> **Table B**
>
> Table B: Comparison with baseline on 11 visual-language benchmark with equitable data volume.
>
> |                     | VQA       | GQA        | VizWiz   | SQAI     | VQAT     | POPE     | MMEP       | MMEC      | MMB      | LLaVAWd  | MM-Vet   |
> | ------------------- | --------- | ---------- | -------- | -------- | -------- | -------- | ---------- | --------- | -------- | -------- | -------- |
> | LLaVA-1.5-7B        | 78.5     | 62.0      | 50.0     | 66.8     | 58.2     | 85.9     | 1510.7     | –         | 64.3     | 49.0     | 30.5     |
> | VILA-7B             | 79.9    | 62.3      | 57.8     | 68.2     | 64.4     | 85.5     | 1533.0     | 296.1     | 68.9     | 70.0     | 34.9     |
> | Sugar               | 76.0    | 58.7     | 60.4     | 69.4     | 57.5     | 86.6     | 1550.8     | 300.0     | 64.9     | 75.6     | 31.3     |
> | with equitable data volume | **80.2** | **63.1** | **61.0** | **72.1** | **65.1** | **87.2** | **1550.0** | **309.0** | **69.3** | **76.4** | **36.8** |
>
> *We also present the results in Table B of the Rebuttal PDF.*

---

> ### Author Response · Authors · 2024-08-07
> **Table D**
>
> Table D: Knowledge-based VQA.
> |                               | FVQA     | WebQA    |
> | ----------------------------- | -------- | -------- |
> | LLaVA-1.5-7B                  | 5.9      | /        |
> | LLaVA-1.5-7B + CLIP image     | 6.8      | 81.8     |
> | LLaVA-1.5-7B + CLIP text      | 7.1      | 79.2     |
> | LLaVA-1.5-7B + CLIP (average) | 7.9      | /        |
> | VILA-7B                       | 6.4      | /        |
> | VILA-7B + CLIP image          | 9.0      | 80.0     |
> | VILA-7B + CLIP text           | 10.2     | 71.2     |
> | VILA-7B + CLIP (average)      | 11.0     | /        |
> | **Sugar**                     | **6.5**  | /        |
> | **Sugar + rag**               | **20.7** | **88.7** |
>
> *We also present the results in Table D of the Rebuttal PDF.*

---

> ### Author Response · Authors · 2024-08-07
> **Table E**
>
> Table E: Comparison with baseline in 5 challenging tasks.
> |                     | **Multi-image VQA** |          | **In-context Interaction** |          | **Visual Prompt** |
> | ------------------- | ------------------- | -------- | -------------------------- | -------- | ------------------ |
> |                     | SEED                | Mantis   | Visual Dialog              | VIST-SIS | BLINK              |
> | LLaVA-1.5-7B        | 58.6                | 31.3     | 53.7                       | 14.1     | 37.1               |
> | VILA-7B             | 61.1                | 38.4     | 58.5                     | 17.4     | 39.2               |
> | Sugar | **63.6**            | **41.0** | **62.5**                 | **28.2** | **42.2**           |
>
> *We also present the results in Table E of the Rebuttal PDF.*

---

> ### Author Response · Authors · 2024-08-10
> **Looking Forward to Your Reply**
>
> Dear Reviewer 4VP3,
>
> Thank you for the time and effort you have dedicated to reviewing our submission. We hope we have addressed the concerns raised in your initial reviews and eagerly await your thoughts and further guidance to refine our work. As the author-reviewer discussion period for NeurIPS 2024 is half past, please let us know if you require any additional information or clarification from our end. We are open to engage in further discussions to enhance our submission.

---

> ### Author Response · Authors · 2024-08-11
> **Awaiting Your Feedback**
>
> Dear Reviewer 4VP3,
>
> Thank you again for reviewing our submission. As the author-reviewer discussion period for NeurIPS 2024 is nearly over, please let us know if any further information or clarification is needed. We are ready to engage in any further discussions with you！
> Looking forward to your further feedback!

---

> > ### Comment · Reviewer_4VP3 · 2024-08-12
> > **Response to rebuttal**
> >
> > Thanks for authors' rebuttal, I think they have addressed most of my concerns, so I will raise my score.

---

> > > ### Author Response · Authors · 2024-08-12
> > > **Official Comment by Authors**
> > >
> > > Thank you for increasing the score. Your valuable suggestions greatly contribute to the quality of our manuscript. Thank you again for your precious time and valuable suggestions!

---

### Author Rebuttal · Authors · 2024-08-07

We sincerely thank all the reviewers for their insightful and valuable comments!

We thank all the reviewers for agreeing that this paper presents a very interesting idea of **addressing the limitations of the original generative paradigm** in comprehensively capturing global information and keenly discerning fine-grained semantic details through the introduction of discriminative supervision. Collaterally, our method leverages the strengths of MLLMs to handle **complex information retrieval** tasks that cannot be effectively solved by discriminative models such as CLIP. Our paradigm also realizes **retrieval-augmented generation** and addresses **knowledge-based** VQA problems within a **single model**, showing great promise for future work.

Overall, we are encouraged that they find that:

- The motivation and novelty for Sugar are clear, reasonable, and meaningful. (all Reviewers)
- Clearly articulate the challenges faced by existing MLLMs and provide a well-defined solution. (all Reviewers)
- The experiments are thorough, clearly showing that the proposed framework can excel well at both comprehension and discrimination tasks. (Reviewers Lcdg and Reviewers DDMF)
- The effect of combining our retriever and generator is interesting, which can inspire fellow works (Reviewer Lcdg).

To address the concerns raised by the reviewers, overall we have conducted several additional experiments and analyses:

- We further validated Sugar's effectiveness using more data equivalent to the baseline, as shown in **Tables B and C of the Rebuttal PDF**.
- We present the performance across 5 tasks of 3 new types (in-context, multi-image, visual prompt comprehension) to more comprehensively evaluate our model's capabilities in **Table E of the Rebuttal PDF**.
- We conducted a more detailed dataset and method ablations to verify the stability of Sugar and the effectiveness of each component in Sugar in **Table G of the Rebuttal PDF**.
- We validated the 2 tasks requiring external knowledge in **Table D of the Rebuttal PDF** to demonstrate the benefit of combining retrieval and comprehension abilities in a single model, thereby avoiding compatibility issues and sub-optimal performance.

Next, we address each reviewer's detailed concerns point by point. We sincerely thank all reviewers for their recognition of our work and the valuable suggestions provided! And discussions are always open. Thank you!

---

### Decision · Program_Chairs · 2024-09-25

**Decision:**

Accept (poster)

**Comment:**

This paper proposes a novel method for unifying generative and discriminative training paradigms in multi-modal large language models (MLLMs). The approach aims to address the limitations of existing Vision-Language Models (VLMs) by integrating generative training, which typically struggles with issues like hallucinations, and discriminative training, which often falls short in complex scenarios requiring fine-grained semantic differentiation. The authors introduce a structure-induced training strategy that leverages dynamic sequence alignment within the Dynamic Time Warping (DTW) framework, enhancing the model's ability to capture both global and fine-grained semantics.
After the rebuttal, this paper received all positive recommendations (two weak accepts and one borderline accept).  The major concern from reviewers was about the Sugar's effectiveness, which has been addressed in the rebuttal. The AC agrees with the reviewer’s assessments and recommends accepting the paper.